# Performance assessment and economic analysis of a human Liver-Chip for predictive toxicology

Lorna Ewart [1✉], Athanasia Apostolou[1], Skyler A. Briggs[1], Christopher V. Carman[1], Jake T. Chaff [1], Anthony R. Heng [1], Sushma Jadalannagari[1], Jeshina Janardhanan[1], Kyung-Jin Jang[1], Sannidhi R. Joshipura[1], Mahika M. Kadam[1], Marianne Kanellias [1], Ville J. Kujala[1], Gauri Kulkarni[1], Christopher Y. Le[1], Carolina Lucchesi[1], Dimitris V. Manatakis [1], Kairav K. Maniar [1], Meaghan E. Quinn[1], Joseph S. Ravan[1], Ann Catherine Rizos[1], John F. K. Sauld[1], Josiah D. Sliz[1], William Tien-Street[1], Dennis Ramos Trinidad[1], James Velez[1], Max Wendell [1], Onyi Irrechukwu[2], Prathap Kumar Mahalingaiah[3], Donald E. Ingber [4,5,6], Jack W. Scannell[7] & Daniel Levner[1]

## Abstract

**Background** Conventional preclinical models often miss drug toxicities, meaning the harm these drugs pose to humans is only realized in clinical trials or when they make it to market. This has caused the pharmaceutical industry to waste considerable time and resources developing drugs destined to fail. Organ-on-a-Chip technology has the potential to improve success in drug development pipelines, as it can recapitulate organ-level pathophysiology and clinical responses; however, systematic and quantitative evaluations of Organ-Chips' predictive value have not yet been reported.

**Methods** 870 Liver-Chips were analyzed to determine their ability to predict drug-induced liver injury caused by small molecules identified as benchmarks by the Innovation and Quality consortium, who has published guidelines defining criteria for qualifying preclinical models. An economic analysis was also performed to measure the value Liver-Chips could offer if they were broadly adopted in supporting toxicity-related decisions as part of preclinical development workflows.

**Results** Here, we show that the Liver-Chip met the qualification guidelines across a blinded set of 27 known hepatotoxic and non-toxic drugs with a sensitivity of 87% and a specificity of 100%. We also show that this level of performance could generate over $3 billion annually for the pharmaceutical industry through increased small-molecule R&D productivity.

**Conclusions** The results of this study show how incorporating predictive Organ-Chips into drug development workflows could substantially improve drug discovery and development, allowing manufacturers to bring safer, more effective medicines to market in less time and at lower costs.

## Plain language summary

Drug development is lengthy and costly, as it relies on laboratory models that fail to predict human reactions to potential drugs. Because of this, toxic drugs sometimes go on to harm humans when they reach clinical trials or once they are in the marketplace. Organ-on-a-Chip technology involves growing cells on small devices to mimic organs of the body, such as the liver. Organ-Chips could potentially help identify toxicities earlier, but there is limited research into how well they predict these effects compared to conventional models. In this study, we analyzed 870 Liver-Chips to determine how well they predict drug-induced liver injury, a common cause of drug failure, and found that Liver-Chips outperformed conventional models. These results suggest that widespread acceptance of Organ-Chips could decrease drug attrition, help minimize harm to patients, and generate billions in revenue for the pharmaceutical industry.

[1] Emulate Inc., 27 Drydock Avenue, Boston, MA, USA. [2] Janssen Pharmaceuticals, Spring House, Philadelphia, PA, USA. [3] Investigative Toxicology and Pathology, Abbvie, North Chicago, IL, USA. [4] Wyss Institute for Biologically Inspired Engineering, Harvard University, Boston, MA, USA. [5] Harvard John A. Paulson School of Engineering and Applied Sciences, Harvard University, Cambridge, MA, USA. [6] Vascular Biology Program and Department of Surgery, Harvard Medical School and Boston Children's Hospital, Boston, MA, USA. [7] JW Scannell Analytics LTD, 32 Queens Crescent, Edinburgh EH9 2BA, UK. ✉email: lorna.ewart@emulatebio.com

Despite billion-dollar investments in research and development, the process of approving new drugs remains lengthy and costly due to high attrition rates[1–3]. Failure is common because the models used preclinically—which include computational, traditional cell culture, and animal models—have limited predictive validity[4]. The resulting damage to productivity in the pharmaceutical industry causes concern across a broad community of drug developers, investors, payers, regulators, and patients, the last of whom desperately need access to medicines with proven efficacy and improved safety profiles. Approximately 75% of the cost in research and development is the cost of failure[5]—that is, money spent on projects in which the candidate drug was deemed efficacious and safe by early testing but was later revealed to be ineffective, unsafe, or otherwise of limited commercial value during human clinical trials. Pharmaceutical companies are addressing this challenge by learning from drugs that failed and devising frameworks to unite research and development organizations to enhance the probability of clinical success[6–9]. One of the major goals of this effort is to develop preclinical models that could enable a "fail early, fail fast" approach, which would result in candidate drugs with greater probability of clinical success, improved patient safety, lower cost, and a faster time to market.

There are important practical challenges in ascertaining the predictive validity of new preclinical models, as there is a broad diversity of chemistries and mechanisms of action or toxicity to consider, as well as considerable time needed to confirm the model's predictions once tested in the clinic. Consequently, arguments for the adoption of these new models are often based on features that are presumed to correlate with human responses to pharmacological interventions—realistic histology, similar genetics, or the use of patient-derived tissues. But even here there is a common problem in much of the academic literature: the important model features are chosen post hoc by the authors and not prospectively by an independent third party that has expertise in the therapeutic problem at hand[10].

The Innovation and Quality (IQ) consortium is a collaboration of pharmaceutical and biotechnology companies that aims to advance science and technology to enhance drug discovery programs. To further this goal, the consortium has described a series of performance criteria that a new preclinical model must meet to become qualified. Within this consortium is an affiliate dedicated to microphysiological systems (MPS), including Organ-on-a-Chip (Organ-Chip) technology, which employs microfluidic engineering to recapitulate in vivo cell and tissue microenvironments in an organ-specific context[11,12]. This is achieved by recreating tissue-tissue interfaces and providing fine control over fluid flow and mechanical forces[13,14], optionally including supporting interactions with immune cells[15] and microbiome[16], and reproducing clinical drug exposure profiles[17]. Recognizing the promise of MPS for drug research and development, the IQ MPS affiliate has provided guidelines for qualifying new models for specific contexts of use to help advance regulatory acceptance and broader industrial adoption[18]; however, to this date, there have been no publications describing studies that carry out this type of performance validation for any specific context of use or that demonstrate an MPS capable of meeting these IQ consortium performance goals.

Guided by the IQ MPS affiliate's roadmap on liver MPS[19], which states that in vitro models for predicting drug-induced liver injury (DILI) that meet its guidelines are more likely to exhibit higher predictive validity than those that do not, we rigorously assessed commercially available human Liver-Chips (from Emulate, Inc.) within the context of use of DILI prediction. In this study, we tested 870 Liver-Chips using a blinded set of 27 different drugs with known hepatotoxic or non-toxic behavior recommended by the IQ consortium (Table 1). We compared the

**Table 1 Small-molecule drugs used in the Liver-Chip evaluation.**

| Drug | IQ MPS list | Tested in spheroid | Spheroid false negative | Garside DILI rank |
|---|---|---|---|---|
| *Ambrisentan* | Yes, matched with Sitaxsentan | Yes | No | 5 |
| Asunaprevir | Yes, no matched pair | No | No | 2 |
| Benoxaprofen | No | Yes | Yes | 1 |
| Beta-Estradiol | No | Yes | Yes | 3 |
| *Buspirone* | Yes, matched with Nefazodone | Yes | No | 4 |
| Chlorpheniramine | No | Yes | Yes | 3 |
| *Clozapine* | Yes, matched with Olanzapine | Yes | No | 2 |
| Diclofenac | Yes, no matched pair | Yes | No | 2 |
| *Entacapone* | Yes, matched with Tolcapone | Yes | No | 4 |
| *Fialuridine* | Yes, matched with FIRU | Yes | No | 1 |
| *FIRU* | Yes, matched with Fialuridine | No | No | 5 |
| Labetalol | No | Yes | Yes | 1 |
| *Levofloxacin* | Yes, matched with Trovafloxacin | Yes | Yes | 2 |
| Lomitapide | No, Mipomersen substitute | No | No | 3 |
| *Nefazodone* | Yes, matched with Buspirone | Yes | No | 1 |
| | Yes, matched with Clozapine | No | No | 5 |
| *Pioglitazone* | Yes, matched with Troglitazone | Yes | Yes | 3 |
| Simvastatin | No | Yes | Yes | 2 |
| *Sitaxsentan* | Yes, matched with Ambrisentan | Yes | No | 1 |
| Stavudine | No | Yes | Yes | 1 |
| Tacrine | No | Yes | Yes | 2 |
| Telithromycin | Yes, no matched pair | No | No | 1 |
| *Tolcapone* | Yes, matched with Entacapone | Yes | No | 1 |
| *Troglitazone* | Yes, matched with Pioglitazone | Yes | No | 1 |
| *Trovafloxacin* | Yes, matched with Levofloxacin | Yes | No | 1 |
| Ximelagatran | No | Yes | Yes | 1 |
| Zileuton | Yes, no matched pair | Yes | Yes | 2 |

The 27 small-molecule drugs are listed according to the IQ MPS affiliate classification and their ranking in the Garside DILI severity category, where 1 corresponds to drugs with severe clinical DILI and 5 to those with no DILI[31,50]. Structurally related toxic and non-toxic pairs are indicated as well using bold, italic text.

results to the historical performance of animal models as well as 3D spheroid cultures of primary human hepatocytes, which are preclinical models frequently employed in this context of use in the pharmaceutical industry[20]. In addition, we analyzed the Liver-Chip results from an economic perspective by estimating the financial value that Liver-Chips could offer if they saw broad adoption in supporting toxicity-related decisions as part of pre-clinical development workflows. Our study found the Liver-Chip to meet the IQ consortium guidelines and confirmed it to be a highly predictive model based on determinations of sensitivity, specificity, and Spearman correlation calculations. We also calculated that routine adoption of the Liver-Chip into preclinical workflows would generate an additional $3 billion annually for the pharmaceutical industry.

## Methods

**Cell culture**. Cryopreserved primary human hepatocytes, purchased from Gibco (Thermo Fisher Scientific), and cryopreserved primary human liver sinusoidal endothelial cells (LSECs), purchased from Cell Systems, were cultured according to their respective vendor/Emulate protocols. The LSECs were expanded at a 1:1 ratio in 10–15 T-75 flasks (Corning) that were pre-treated with 5 mL of Attachment Factor (Cell Systems). Complete LSEC medium includes Cell Systems medium with final concentrations of 1% Pen/Strep (Sigma), 2% Culture-Boost (Cell Systems), and 10% Fetal Bovine Serum (FBS) (Sigma). Media was refreshed daily until cells were ready for use. Cryopreserved human Kupffer cells (Samsara Sciences) and human Stellate cells (IXCells) were thawed according to their respective vendor/Emulate protocols on the day of seeding. See Supplementary Table 1 for further information.

**Liver-Chip microfabrication and Zoë® culture module**. Each chip (Fig. 1) is made from flexible polydimethylsiloxane (PDMS), a transparent viscoelastic polymer material. The chip compartmental chambers consist of two parallel microchannels that are separated by a porous membrane containing pores of 7 μm diameter spaced 40 μm apart.

On Day −6, chips were functionalized using Emulate proprietary reagents, ER-1 (Emulate reagent: 10461) and ER-2 (Emulate reagent: 10462), mixed at a concentration of 1 mg/mL prior to application to the microfluidic channels of the chip. The platform is then irradiated with high power UV light

(peak wavelength: 365 nm, intensity: 100 μJ/cm$^2$) for 20 minutes using a UV oven (CL-1000 Ultraviolet Crosslinker AnalytiK-Jena: 95-0228-01). Chips were then coated with 100 μg/mL Collagen I (Corning) and 25 μg/mL Fibronectin (ThermoFisher) in both channels. The top channel was seeded with primary human hepatocytes on Day −5 at a density of $3.5 \times 10^6$ cells/mL. Complete hepatocyte seeding medium contains Williams' Medium E (Sigma) with final concentrations of 1% Pen/Strep (Sigma), 1% L-GlutaMAX (Gibco), 1% Insulin-Transferring-Selenium (Gibco), 0.05 mg/mL Ascorbic Acid (Sigma), 1 μM dexamethasone (Sigma), and 5% FBS (Sigma). After four hours of attachment, the chips were washed by gravitational force. Gravity wash consisted of gently pipetting 200 μL of fresh medium at the top inlet, allowing it to flow through, washing out any unbound cells from the surface, and inserting a pipette tip on the outlet of the channel.

On Day −4, a hepatocyte Matrigel overlay procedure was executed with the purpose of promoting a three-dimensional matrix for the hepatocytes to grow in an ECM sandwich culture. The hepatocyte overlay and maintenance medium contains Williams' Medium E (Sigma) with final concentrations of 1% Pen/Strep (Sigma), 1% L-GlutaMAX (Gibco), 1% Insulin-Transferrin-Selenium (Gibco), 50 μg/mL Ascorbic Acid (Sigma), and 100 nM Dexamethasone (Sigma). On Day −3, the bottom channel was seeded with LSECs, stellate cells and Kupffer cells, further known as non-parenchymal cells (NPCs). NPC seeding medium contains Williams' Medium E (Sigma) with final concentrations of 1% Pen/Strep (Sigma), 1% L-GlutaMAX (Gibco), 1% Insulin-Transferrin-Selenium (Gibco), 50 μg/mL Ascorbic Acid (Sigma), and 10% FBS (Sigma). LSECs were detached from flasks using Trypsin (Sigma) and collected accordingly. These cells were seeded in a mixture volume ratio of 1:1:1 with LSECs at a density range of $9–12 \times 10^6$ cells/mL, stellates at a density of $0.3 \times 10^6$ cells/mL, and Kupffer cells at $6 \times 10^6$ cells/mL followed by a gravity wash 4 h post-seeding.

On Day −2, chips were visually inspected under the ECHO microscope (Discover Echo, Inc.) for cellular maturation and attachment, healthy morphology, and a tight monolayer. The chips that passed visual inspection had both channels washed with their respective media, leaving a droplet on top. NPC maintenance media was composed of the same components prior, with a reduction of FBS to 2%. To minimize bubbles within the system, one liter of complete, warmed top and bottom media was added to Steriflip-connected tubes (Millipore) in the biosafety

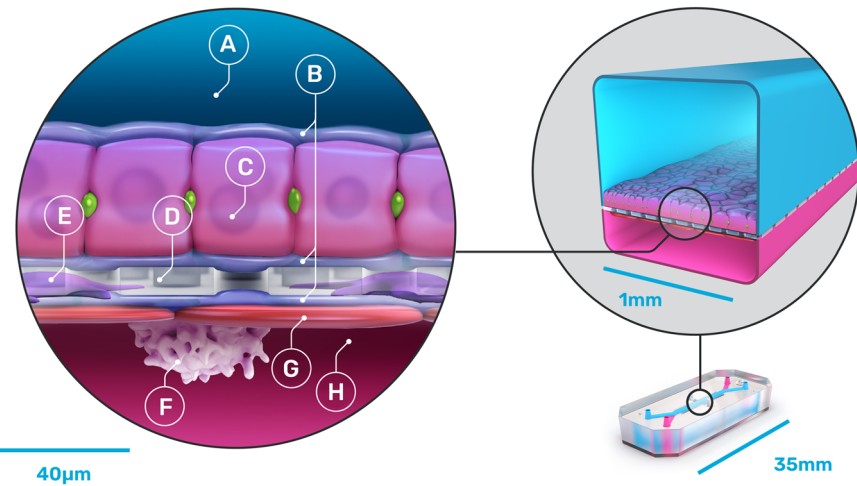

**Fig. 1 Schematic of the Emulate Liver-Chip.** This diagram shows primary human hepatocytes (C) that are sandwiched within an extracellular matrix (B) on a porous membrane (D) within the upper parenchymal channel (A), while human liver sinusoidal endothelial cells (G), Kupffer cells (F), and stellate cells (E) are cultured on the opposite side of the membrane in the lower vascular channel (H).

cabinet. All media was then degassed using a $-70$ kPa vacuum (Welch) and stored in the incubator until use. Pods were primed twice with 3 mL of degassed media in both inlets and 200 µL in both outlets. Chips were then connected to pods via liquid-to-liquid connection. Chips and pods were placed in Zoë® (Emulate Inc.) for their first Regulate Cycle, which minimizes bubbles within the fluidic system by increasing the pressure for two hours. After this, normal flow resumed at 30 µL/h. On Day $-1$, Zoë® was set to regulate once more.

**Experimental setup**. The 870-chip experiment was carried out in five consecutive cycles (herein referred to as Cycles 1 through 5) to test a selection of 27 drugs at varying concentrations relative to the average therapeutic human $C_{max}$ obtained from literature (Supplementary Data 1). Cycles 1 to 4 tested 6–8 concentrations in duplicate for each of 10–13 drugs. To determine which sampling strategy was optimal for cycles 1 to 4 (16 doses × 1 replicate, 8 × 2, 5 × 3, or 4 × 4), we generated three different "dose-response" synthetic datasets, each distorted by different noise levels (low, medium, or high). For each of these datasets, we performed curve-fitting analyses and calculated the Root Mean Square Error between the "true" and estimated $IC_{50}$ parameter. The analysis results showed that, for all noise levels, the 16 × 1 sampling strategy marginally outperformed the 8 × 2. However, to ensure at least two replicates per concentration, the 8 × 2 strategy was selected. Some drugs were also repeated across cycles (on different donors or at different concentrations) to ensure experimental robustness. However, to replicate a more typical study design likely to be carried out by scientists in the pharmaceutical industry, we created cycle 5 with 6 drugs, where each was tested in 4 concentrations in triplicate. For each cycle, chips were dosed with drug over 8 days (referred to as Day 0 through Day 7). Drug preparation, dosing, and analysis teams were divided, creating a double-blind study such that those administering the drugs or performing analyses did not know the name or concentrations of the drugs tested.

**Drug preparation**. The drug dosing concentrations were determined from the unbound human $C_{max}$ of each drug. First, the expected fraction of drug unbound in media with 2% FBS was extrapolated from plasma binding data for each drug. Dosing concentrations were then back calculated such that the unbound fraction in media would reflect relevant multiples of unbound human $C_{max}$ (Supplementary Data 1). For each cycle, concentrations ranged from 0.1 to 1000 times $C_{max}$.

Stock solutions were prepared at 1000 times the final dosing concentration. Drugs in powder form were either weighed out with 1 mg precision or dissolved directly in vendor-provided vials. Sterile DMSO (Sigma) was added to dissolve the drug. The solution was triturated before transferring to an amber vial (Qorpak), which was vortexed (Fisher Scientific) on high for 60 s to ensure complete dissolution. A serial dilution was then performed in DMSO to prepare each subsequent 1000× concentration. These stock solutions were then aliquoted in 1.5 mL tubes (Eppendorf) and stored at $-20$ °C until dosing day, allowing a maximum of one freeze-thaw cycle prior to dosing.

All media was made the day prior to chip dosing and stored overnight at 37 °C. On the day of chip dosing, one stock aliquot per drug concentration was thawed in a 37 °C bead bath. The stocks were then vortexed and inspected to ensure absence of drug particulate. Dosing solutions were prepared by diluting drug stock 1:1000 in top or bottom media to achieve 0.1% DMSO concentration. The dosing solutions were then vortexed and stored at 37 °C until dosing.

On the first dosing day (Day 0), all chips were imaged using the ECHO microscope. Five hundred microliters of effluent was collected from all four reservoirs of the pod and placed in a labeled 96-well plate. After collection, all the remaining media was carefully aspirated before dosing with 3.8 mL of corresponding dosing solution. Dosing occurred on study days 0, 2, and 4 for chips flowing at 30 µL/h, and on days 0, 1, 2, 3, 4, 5, and 6 for chips flowing at 150 µL/h. Effluent collection occurred on days 1, 3, and 7.

**Biochemical assays**. Top channel outlet effluents were analyzed to quantify albumin and alanine transaminase (ALT) levels on days 1, 3, and 7 using sandwich ELISA kits (Abcam, Albumin ab179887, ALT ab234578) according to vendor-provided protocols. Frozen ($-80$ °C) effluent samples were thawed overnight at 4 °C prior to assay. The Hamilton Vantage liquid handling platform was used to manage effluent dilutions (1:500 for albumin, neat for ALT), preparation of standard curves, and addition of antibody cocktail. Absorbance at 450 nm was measured using the SynergyNeo Microplate Reader (BioTek).

As part of cycles 3, 4, and 5, top channel outlet samples from vehicle chips on days 1, 3, and 7 post-drug or vehicle administration were analyzed to quantify urea levels with a urea assay kit (Sigma–Aldrich, MAK006) according to vendor-provided protocol. Frozen ($-80$ °C) effluent samples were thawed overnight at 4 °C prior to assay. All samples were diluted 1:5 in assay buffer and mixed with the kit's Reaction Mix. Absorbance at 570 nm was measured using the same automated plate reader.

Effluent samples from vehicle chips and those treated with either trovafloxacin or levofloxacin were thawed overnight at 4 °C, and effluents from both channels were analyzed for IL-6 and TNF-alpha levels using MSD U-PLEX kits (Meso Scale Diagnostics, K15067L-2) according to vendor-provided protocols. Samples were added to plates manually at a 1:2 dilution. Plates were read for cytokine release on the MESO QuickPlex SQ 120 (Meso Scale Discoveries).

**Morphological analysis**. At least four to six brightfield images were acquired per chip for morphology analysis. Brightfield images were acquired on the ECHO microscope using these settings: 170% zoomed phase contrast, 50% LED, 38% brightness, 41% contrast, 50% color balance, color on, and ×10 objective. Brightfield images were acquired across three fields of view on days 1, 3, and 7 for each cycle. Cytotoxicity classification was performed while acquiring images for both NPCs and hepatocytes. Images were then scored zero to four by blinded individuals ($n = 2$) based on severity of agglomeration of cell debris for both channels. The scoring matrix and representative images have been included in the supplement (Supplementary Fig. 1).

At the end of the experiment, cells in the Liver-Chip were fixed using 4% paraformaldehyde (PFA) solution (Electron Microscopy Sciences). Chips were detached from pods and washed once with PBS. The PFA solution was pipetted into both channels and incubated for 20 min at room temperature. Afterwards, chips were washed with PBS and stored at 4 °C until staining. Following fixation, chips corresponding to low, medium, and high concentrations from each group were cut in half with a razor blade perpendicular to the co-culture channels. One half was used in the following staining protocol, while the other half was stored for future staining. All stains and washes utilized the bubble method, in which a small amount of air is flowed through the channel prior to bulk wash media to prevent a liquid-liquid dilution of the staining solution. The top channel was perfused with 100 µL of AdipoRed (Lonza, PT-7009) diluted 1:40 v/v in PBS labeling lipid accumulation and 100 µL of NucBlue

(ThermoFisher, R37605) (100 drops in 50 mL of PBS) to visualize cell nuclei. Following 15 minutes of incubation at room temperature, each channel was washed with 200 μL of PBS (alternating channels, 2× for top and 3× for bottom). As an alternative lipid accumulation marker, 100 μL of HCS LipidTOX Deep Red (ThermoFisher, H34477) was diluted 1:1000 v/v in PBS with NucBlue counterstain and added to the top channel. After a 30 min incubation at room temperature, the chips were again allowed to reach room temperature before the channels were alternatively washed with 200 μL of PBS, the bottom channel undergoing three washes and the top channel undergoing two. Chips were then imaged using the Opera Phenix.

Following lipid and DAPI staining and imaging, chips were stained with a multi-compound resistant protein 2 (MRP2) antibody to visualize the bile canalicular structures characteristic of healthy Liver-Chips. First, chips were permeabilized in 0.125% Triton-X and 2% Normal Donkey Serum (NDS) diluted in PBS (100 μL of solution per channel) and incubated at room temperature for 10 min. Then, each channel was washed with 200 μL of PBS (alternating channels, 2× for top and 3× for bottom). Chips were then blocked in 2% Bovine Serum Albumin (BSA) and 10% NDS in PBS (100 μL of solution per channel) and incubated at room temperature for 1 h. Next, primary antibody Mouse anti-MRP2 (Abcam, ab3373) was prepared 1:100 in the original blocking buffer, diluted 1:4 in PBS. 100 μL of solution was added to each channel, and chips were stored overnight at 4 °C. The following day, each channel was washed with 200 μL of PBS (alternating channels, 2× for top and 3× for bottom). Secondary antibody Donkey anti-Mouse 647 (Abcam, ab150107) was prepared 1:500 in original blocking buffer, diluted 1:4 in PBS. 100 μL of solution was added to each channel and chips incubated at room temperature, protected from light, for two hours. Then, each channel was washed with 200 μL of PBS (alternating channels, 2× for top and 3× for bottom). Chips were imaged immediately or stored at 4 °C until ready for imaging on the Opera Phenix.

**Live staining**. Chip replicates designated for live cell imaging were washed with PBS utilizing the bubble method. Chips were then cut in half perpendicular to the co-culture channels. The top chip halves were stained with NucBlue (ThermoFisher, R37605) to visualize cell nuclei and Cell Event Green (ThermoFisher, C10423) to visualize activated caspase 3/7 for apoptosis. This staining panel was prepared in serum-free media (CSC), with NucBlue at 2 drops per mL and Cell Event Green at a 1:500 ratio and perfused through both channels. The bottom chips halves were stained with NucBlue (Thermo) to visualize nuclei and Tetramethylrhodamine, methyl ester (TMRM) (ThermoFisher, I34361) to visualize active mitochondria. This staining panel was prepared in PBS with 5% FBS, with NucBlue at 2 drops per mL and TMRM at a 1:1000 ratio in original blocking buffer, diluted 1:4 in PBS. Chips were incubated in the dark at 37 °C for 30 min, and then each channel was washed with 200 μL of PBS (alternating channels, 2× for top and 3× for bottom). The chips were kept at 37 °C, protected from light, until ready for imaging with the Opera Phenix.

**Image acquisition**. Fluorescent confocal image acquisition was performed using the Opera Phenix High-Content Screening System and Harmony 4.9 Imaging and Analysis Software (PerkinElmer). Before acquisition, the Phenix internal environment was set to 37 °C and 5% $CO_2$. Chips designated for imaging were removed from their plates, wiped on the bottom surface to remove moisture, and placed into the Phenix 12-chip imaging adapter. Whole chips were placed directly into each slot, while top and bottom half chips

were matched and combined in one chip slot. Chips were aligned flush with the adapter and one another. Any bubbles identified from visual inspection were washed out with PBS. Once ready, the stained chips were covered with transparent plate film to seal channel ports and loaded into the Phenix. For live imaging, the DAPI (Time: 200 ms, Power: 100%), Alexa 488 (Time: 100 ms, Power: 100%), and TRITC (Time: 100 ms, Power: 100%) lasers were used. For fixed imaging, the DAPI (Time: 200 ms, Power: 100%), TRITC (Time: 100 ms, Power: 100%), and Alexa 647 (Time: 300 ms, Power: 80%) lasers were used. Z-stacks were generated with 3.6 μm between slices for 28–32 planes so that the epithelium was located around the center of the stack. Six fields of view (FOVs) per chip were acquired, with a 5% overlap between adjacent FOVs to generate a global overlay view.

**Image analysis**. Raw images from fixed and live imaging were exported in TIFF format from the Harmony software. Using scripts written for FIJI (ImageJ), TIFFs across three color channels and multiple z-stacks were compiled into composite images for each field of view in each chip. The epithelial signal was identified and isolated from the endothelial and membrane signals, and the composite TIFFs were split accordingly. The ideal threshold intensity for each channel in the epithelial "substack" was identified to maximize signal, and the TIFFs were exported as JPEGs for further analysis.

**Gene expression analysis**. RNA was extracted from chips using TRI Reagent (Sigma–Aldrich) according to the manufacturer's guidelines. The collected samples were submitted to GENEWIZ (South Plainfield, NJ) for next-generation sequencing. After quality control and RNA-seq library preparation, the samples were sequenced with Illumina HiSeq $2 \times 150$ system using sequencing depth ~50 M paired-ends reads/sample. Using Trimmomatic v0.36, the sequence reads were trimmed to filter out all poor-quality nucleotides and possible adapter sequences. The remaining trimmed reads were mapped to the Homo sapiens reference genome GRCh38 using the STAR aligner v2.5.2b. Next, using the generated BAM files for each sample, the unique gene hit counts were calculated from the Subread package v1.5.2. It is worth noting that only unique reads within the exon region were counted.

**Statistical analysis**. All statistical analyses were conducted in R[21] (version 4.1.2) and figures were produced using the R package ggplot2[22] (version 3.3.5). The dose-response analysis (Fig. 4) was carried out using the popular drc R package developed by Ritz et al.[23] using the generalized log-logistic dose-response model. The error bars in Figs. 3 and 4, correspond to the standard errors of the mean. The circles in Fig. 3 correspond to the samples used to calculate the corresponding statistics. The analysis of significance in Fig. 3 was performed using paired t-test across different days (same donor) and unpaired t-test across different donors (same day). In Fig. 3 the number of samples used were $n = 3$ for donor two, and $n = 4$ for donor three. For both donors, the number of the freshly thawed hepatocyte samples used to estimate the corresponding log2(Fold Change) were $n = 4$. Finally, the analysis of significance in Fig. 3 was performed using paired t-test.

**Economic modeling approach**. An economic model was built to assess the impact of improvements in the predictive validity of preclinical toxicology models on the economics of drug development. This model is provided in full as part of the supplementary materials as a formula-driven Microsoft Excel file (Supplementary Data 2). The model was built by extending the

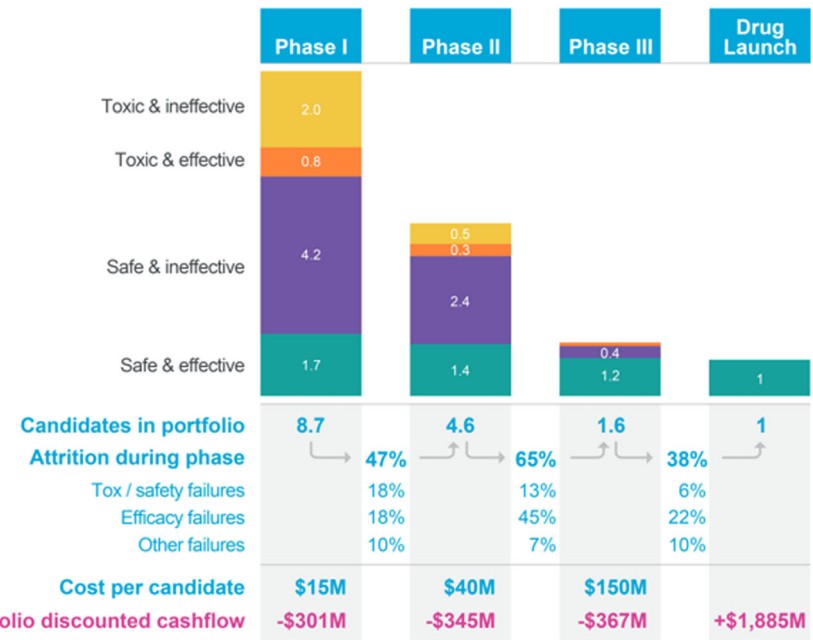

**Fig. 2 Economic value model for assessing the financial impact of improved preclinical testing.** Illustrated is the model's "base case", which tracks a representative portfolio of candidate drugs as it progresses and erodes through clinical trials, culminating in a single drug approval. The model bases phase-by-phase attrition rates ("attrition during phase"), discovery and preclinical costs, development costs ("cost per candidate") and cost of capital on Paul et al.[5] to compute a portfolio-wide discounted cashflow. In contrast with prior approaches, the model tracks the underlying causes of clinical trial failure (safety-related, efficacy-related, and other failures) using parameters derived from literature[7,9,24,25], a feature that permits us to determine the composition of the drug portfolio in each stage of development in terms of candidates that are safe and effective, safe and ineffective, unsafe and effective, and unsafe and ineffective, as illustrated. Improvements in the predictive validity of preclinical safety testing can be captured through their impact on the makeup of the portfolio entering Phase I clinical trials: better preclinical safety testing reduces the proportion of unsafe drugs that enter the clinic relative to the "base case"; the model permits analyzing the impact of such changes on the discounted cashflow and the portfolio's profitability. The model is provided in full in Supplementary Data 2 as a formula-driven Microsoft Excel file.

pipeline model of Paul et al.[5], which tracks the economics of a representative portfolio of candidate drugs as it progresses and erodes through clinical trials. However, in contrast with conventional models, we followed Scannell & Bosley's[4] approach by modeling attrition as a function of decision quality and candidate quality at each development stage. We modeled safety-related failures, efficacy-related failures, and other failures (e.g., commercial and strategy related) with parameters derived from the literature (primarily from Harrison 2016,[7,9,24,25]). The model comprises a "base case", which describes an archetypical drug-development portfolio that leads to a single drug approval. Development costs, timing, cost of capital and attrition rates were set in line with Paul et al.[5]. The base case and its parameters are summarized in Fig. 2.

An innovative feature of the economic model is that it permits us to determine the makeup of the drug portfolio in each stage of development in terms of candidates that are safe and effective, safe and ineffective, unsafe and effective, and unsafe and ineffective. Additionally, the model's structure also allows one to estimate decision quality parameters at each stage of the process, such as the false negative rate (FNR) of the toxicity determination – the proportion of toxic drugs erroneously deemed safe. This, in turn, allows one to estimate the financial impact of changes in predictive validity, something that cannot be done directly with conventional attrition-driven pipeline models.

Improvements in the predictive validity of preclinical safety testing can be captured through their effects on the makeup of the portfolio entering Phase I clinical trials: better preclinical safety testing reduces the proportion of unsafe drugs that enter the clinic. Such improvements are captured by reducing the FNR for

the toxicology testing that occurs between preclinical development and Phase I trials (we also add Organ-Chip costs to capture the price of added testing). If we keep all other model parameters unchanged, the model captures a cost-avoidance strategy: an approach wherein the ability to predict certain clinical trial failures in advance allows one to start fewer clinical trials (skipping those trials that are bound to fail) to bring one drug to market, as in the base case. However, the ability to have more predictable clinical outcomes is not likely to reduce investment but rather to increase it. This increase in R&D productivity should therefore result at least in maintenance of the investment in clinical testing (if not in its increase), which we conservatively model by setting the number of projects entering Phase I to its base case value.

To derive the economic implications of this scenario analysis, we calculate the portfolio's new net present value (NPV) and evaluate its percentage increase (uplift) over the base case. This NPV uplift represents the increase in R&D productivity caused by the improved preclinical testing, which is partially offset by the added cost of Organ-Chip experiments as well as higher clinical trial costs resulting from candidates progressing further in their clinical testing. We estimated the cost of Organ-Chip testing based on Clinical Research Organization (CRO) pricing, to capture direct costs, indirect costs, and third-party profits: this represents a top-end estimate, as pharmaceutical companies may elect to purchase Organ-Chip instrumentation for internal use, leading to lower costs. The model proceeds to apply the NPV uplift to the world-wide R&D spending on small-molecule drug development to estimate the annual financial impact that the increase in R&D productivity may generate.

Because the model is parameterized using historical estimates of attrition rates and their causes, we sought to understand the model's sensitivity to the exact parameter values. To do this, we performed a mathematical sensitivity analysis for the major input parameters; this analysis is included within the Excel file in Supplementary Data 2. The analysis demonstrated that reasonable changes in parameter choices retain the model's qualitative conclusions. This is in part because the model's output is a percentage uplift relative to the base case, making the model robust in the face of uncertainty in the financials of the base case.

**Reporting summary**. Further information on research design is available in the Nature Research Reporting Summary linked to this article.

## Results

**Liver-Chip satisfies IQ MPS affiliate guidelines**. The IQ guidelines for assessment of an in vitro liver MPS within the DILI prediction context of use requires evidence that the model replicates key histological structures and functions of the liver; furthermore, the model must be able to distinguish between 7 pairs of small-molecule toxic drugs and their non-toxic structural analogs. If the model passes through these hurdles, it must demonstrate its ability to predict the clinical responses of 6 additional selected drugs.

The Liver-Chips that we evaluated against these standards contain two parallel microfluidic channels separated by a porous membrane. Following the manufacturer's instructions, primary human hepatocytes are cultured between two layers of extracellular matrix (ECM) in the upper 'parenchymal' channel, while primary human liver sinusoidal endothelial cells (LSECs), Kupffer cells, and stellate cells are placed in the lower 'vascular' channel in ratios that approximate those observed in vivo (Fig. 3). All cells passed quality control criteria that included post-thaw viability > 90%, low passage number (preferably P3 or less), and expression of cell-specific markers. Similar results were obtained using hepatocytes from three different human donors, which were procured from the same commercial vendor (Supplementary Table 1).

Live microscopy of the Liver-Chips revealed a continuous monolayer of hepatocytes displaying cuboidal and binucleated morphology in the upper 'parenchymal' channel of the chips, as well as a monolayer of polygonal shaped LSECs in the bottom 'vascular' channel, on the opposite side of the porous membrane (Fig. 3). Confocal fluorescence microscopy also confirmed liver-specific morphological structures as indicated by the presence of differentiation markers, including bile canaliculi containing a polarized distribution of F-actin and multidrug resistance-associated protein 2 (MRP2; Fig. 3), hepatocytes rich with mitochondrial membrane ATP synthase beta subunit (ATPB; Fig. 3), PECAM-1 (CD31) expressing LSECs, CD68[+] Kupffer cells, and desmin-containing stellate cells (Fig. 3). In addition, transmission electron microscopy confirmed the existence of similar cell-cell relationships and structures to those found in human liver, including well-developed junction-lined bile canaliculi and adhesions between Kupffer cells and sinusoidal endothelial cells (Supplementary Fig. 2).

Albumin and urea production are widely accepted as functional markers for cultured hepatocytes with the goal of reaching computed production levels observed in human liver in vivo (~20–105 μg and 56–159 μg per $10^6$ hepatocytes per day, respectively)[19,26]. Liver-Chips fabricated with cells from three different hepatocyte donors were able to maintain physiologically relevant levels of albumin and urea synthesis over 1 week in culture (Fig. 3, Supplementary Data 3–5). Importantly, in line with the IQ MPS guidelines, the coefficient of variation for the mean daily production rate of urea was always below 5% in all donors on day 1 but increased to 20% on day 7; however, it was higher for albumin production across all donors on day 1 but was between 14 to 27% by day 7. These data corroborate the reproducibility and robustness of the Liver-Chip across experiments and highlight variability across donors that is not unlike the variability observed in humans. In fact, it is important to be able to analyze and understand donor-to-donor variability when evaluating cell-based platforms for the prediction of clinical outcome[27] or when a drug moves into clinical studies.

Because hepatocytes maintained in conventional static cultures rapidly reduce transcription of relevant liver-specific genes[28], the IQ MPS guidelines require confirmation that the genes representing major Phase I and II metabolizing enzymes, as well as uptake and efflux drug transporters, are expressed and that their levels of expression are stable. On days 3 and 7 post-vehicle administration, compared to freshly thawed hepatocytes, we detected high levels of expression in both donors for 13 of the 17 genes requested by IQ MPS, confirming that the chip provides a suitable microenvironment to maintain hepatocytes. Gene expression was notably lower on day 7 compared to day 3 for CYP2D6, CYP2C8, CYP2E1, and MRP2 in donor two and only for MRP2 in donor 3 (Fig. 3, Supplementary Data 6–7). Gene expression levels were lower than freshly thawed hepatocytes for genes encoding OATP1B3, GSTA1, CYP2E1, and CYP2D6, a profile reflected in two donors. Moreover, the demonstration that CY2C9 and CYP3A4 gene expression is maintained above freshly thawed hepatocytes for 7 days post-vehicle or drug administration is encouraging as together the CYP2C and CYP3A families make up 50% of the total CYP population[29]. CYP3A4 is also the major enzyme that metabolizes many marketed drugs. We did not directly assess CYP functional activity in this study but previously, the same Liver-Chip has been shown to exhibit Phase I and II functional activities that are comparable to freshly isolated human hepatocytes and 3D hepatic spheroids[26,30] as well as superior activity relative to hepatocytes in a 2D sandwich-assay plate configuration[26]. Taken together, these data support the notion that the Liver-Chip provides a good microenvironment for hepatocytes to maintain functionality.

As these data confirmed that the Liver-Chip meets the major structural characterization and basic functionality requirements stipulated by the IQ MPS guidelines, we then carried out studies to evaluate this human model as a tool for DILI prediction. IQ MPS identified 7 pairs of small-molecule drugs where one drug has been reported to produce DILI in clinical studies and their structural analog was inactive or exhibited a lower activity and did not produce clinical DILI (Table 1). Past work in the MPS field has focused on technically accessible endpoints that can be easily measured but are unfortunately not clinically relevant or translatable (e.g., $IC_{50}$ for reduction in total ATP content)[31,32]. Furthermore, although cytotoxicity measures are fundamental in the assessment of a drug's potential for hepatotoxicity in vitro[33,34], gene expression and various phenotypic changes can occur at much lower concentrations[35,36]. As the Liver-Chip enables multiple measures of drug effects and use of multiple measures may provide further sensitivity and add value[37], we assessed drug toxicities on days 1, 3, and 7 post-drug or vehicle administration by quantifying both inhibition of albumin production as a general measure of hepatocellular functionality and increases in release of alanine aminotransferase (ALT) protein, which is used clinically as a measure of liver damage. We also scored hepatocyte injury using morphological analysis at 1, 3, and 7 days after drug or vehicle exposure, where higher injury scores indicated greater cellular injury.

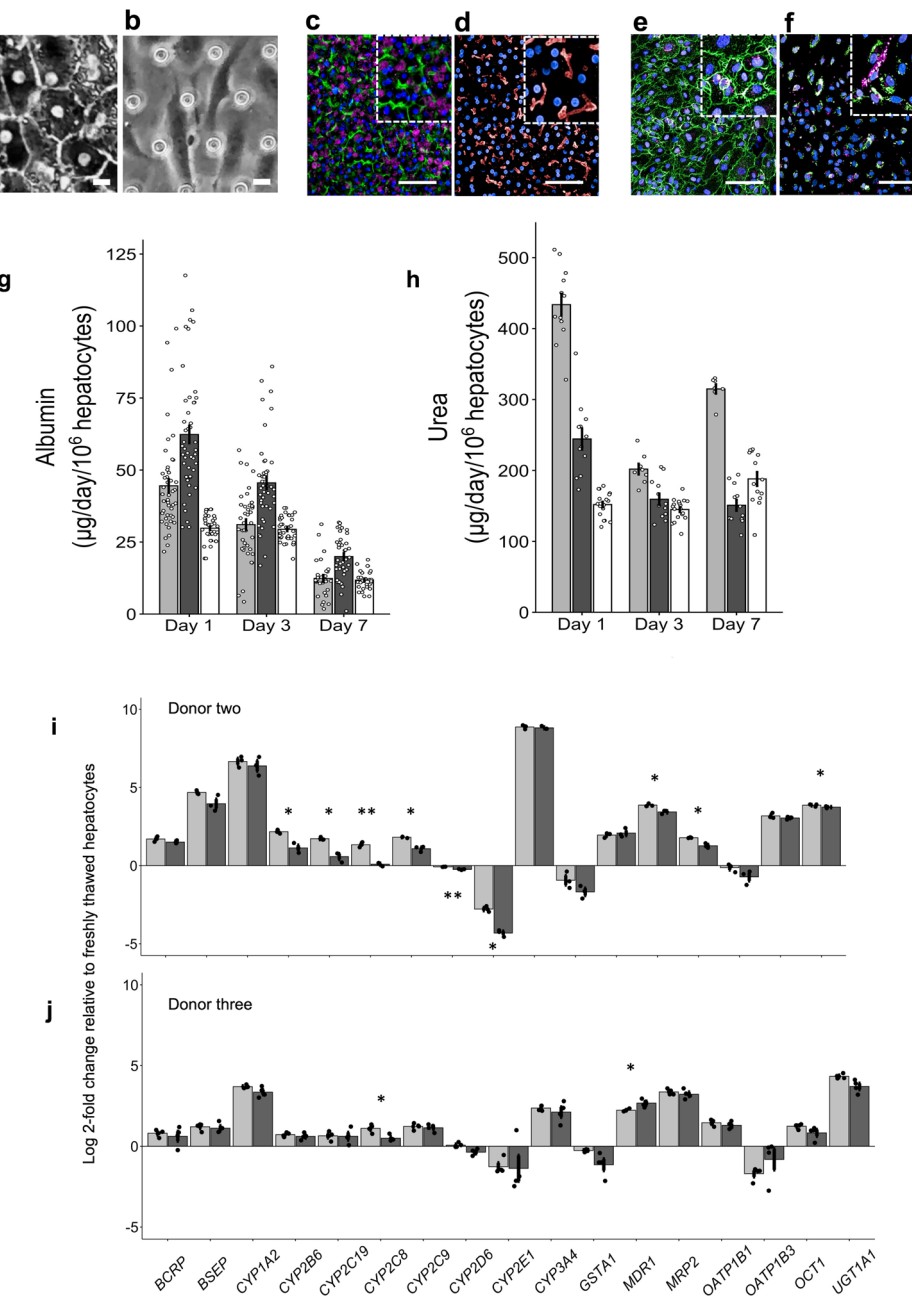

**Fig. 3 Recapitulation of human liver structure and function in the Liver-Chip.** Representative phase contrast microscopic images (scale bar represents 10 μm) of hepatocytes in the upper channel of Liver-Chip (**a**) and non-parenchymal cells in the lower vascular channel (the regular array of circles are the pores in the membrane) (**b**). Representative immunofluorescence microscopic images showing the phalloidin stained actin cytoskeleton (green) and ATPB containing mitochondria (magenta) (**c**) and MRP2-containing bile canaliculi (red) (**d**). CD31-stained liver sinusoidal endothelial cells (green) and desmin-containing stellate cells (magenta) (**e**), and CD68[+] Kupffer cells (green) co-localized with desmin-containing stellate cells (magenta) (**f**). All images in **c**–**f** show DAPI-stained nuclei (blue) and the scale bar represents 100 μm with the inset at 5 times higher magnification; albumin (**g**) and urea (**h**) levels in the effluent from the upper channels of vehicle-treated Liver-Chips created with cells from 3 different donors (light and dark gray bars represent donor one and two respectively, white bars represent donor three) on days 1, 3, and 7 post-vehicle administration, measured by ELISA. Data are presented as mean ± standard error of the mean (S.E.M.). For each condition (i.e., specific donor and day), the exact number of samples used to derive the statistics is: (i) Albumin: donor 1-day 1 ($n = 46$), donor 2-day 1 ($n = 46$), donor 3-day 1 ($n = 39$), donor 1-day 3 ($n = 40$), donor 2-day 3 ($n = 44$), donor 3-day 3 ($n = 38$), donor 1-day 7 ($n = 29$), donor 2-day 7 ($n = 38$), donor 3-day 7 ($n = 30$); (ii) Urea: donor 1-day 1 ($n = 12$), donor 2-day 1 ($n = 12$), donor 3-day 1 ($n = 18$), donor 1-day 3 ($n = 8$), donor 2-day 3 ($n = 12$), donor 3-day 3 ($n = 18$), donor 1-day 7 ($n = 7$), donor 2-day 7 ($n = 12$), donor 3-day 7 ($n = 14$). Levels of key liver-specific genes in control Liver-Chips as determined by RNA-seq analysis on days 3 (light gray) and 7 (dark gray) post-vehicle administration with donor two (**i**) and donor three (**j**). Data are presented as mean Log2 (fold change) ± standard error of the TPM (Transcript Per Million) expression relative to the mean expression of the freshly thawed hepatocytes with $n = 4$ chips; statistical significance of values between day 3 and 7 was determined using a paired $t$-test; *$p < 0.05$, **$p < 0.01$. For each time point (e.g., day 3 and 7), the sample size used to derive the statistics was $n = 3$ for donor two and $n = 4$ for donor three.

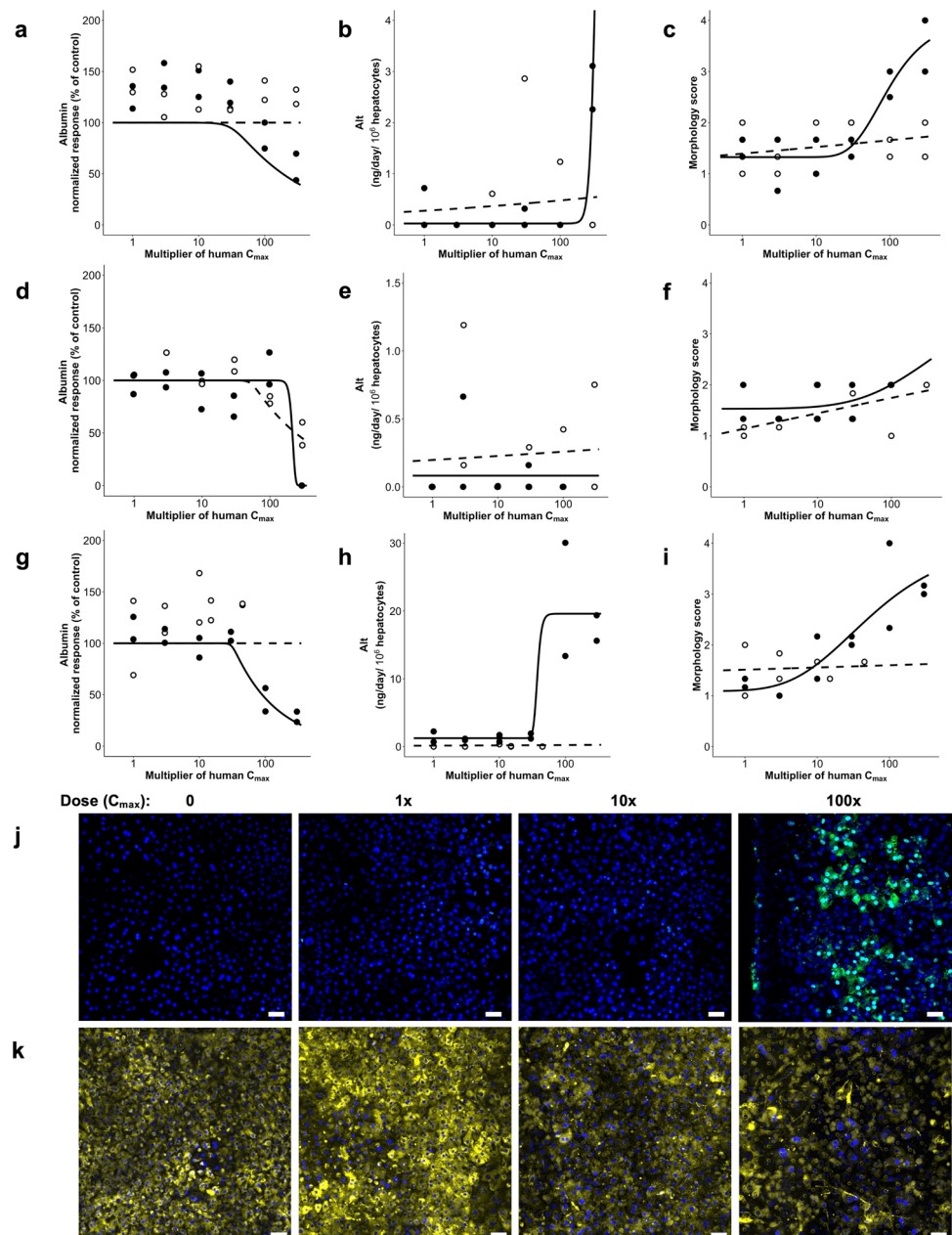

**Fig. 4 Detection of drug concentration-dependent toxicity and liver injury.** Effect of Cloazpine (closed circles) or olanzapine (open circles) on albumin production (**a**), ALT release (**b**), and morphology score (**c**); Effect of troglitazone (closed circles) or pioglitazone (open circles) on albumin production (**d**), ALT release (**e**), and morphology score (**f**); Effect of trovafloxacin (closed circles) or levofloxacin (open circles) on albumin production (**g**), ALT release (**h**) and morphology score (**i**); Immunofluorescence microscopic images showing concentration-dependent increases in caspase 3/7 staining (green;) indicative of apoptosis after treatment with trovafloxacin at 0,1, 10, and 100 (**j**) times the unbound human $C_{max}$ for 7 days; concentration-dependent decrease in TMRM staining (yellow) indicative of mitotoxicity in response to treatment with sitaxsentan at 0,1,10, and 100 (**k**) times the unbound human $C_{max}$ for 7 days. Scale bar represents 50 μm.

We tested the 7 toxic drugs across 8 concentrations that bracket the human plasma $C_{max}$ for each drug based on free (non-protein bound) drug concentrations, with the highest concentrations at 300× $C_{max}$ (unless not permitted by solubility limits as was found for levofloxacin) to represent clinically relevant test concentrations for in vitro models[38] (Supplementary Data 1). The known toxic compounds showed clear concentration- and time-dependent patterns that varied depending on compound. Typically, when albumin production was inhibited, morphological injury scores and ALT levels also increased, but we found that a decrease of albumin production was the most sensitive marker of hepatocyte toxicity in the Liver-Chip, as

shown in sample paired comparisons of clozapine and olanzapine, troglitazone and pioglitazone, and trovafloxacin and levofloxacin (Fig. 4, Supplementary Data 8). Importantly, all 7 of the toxic drugs reduced albumin production or resulted in an increase in ALT protein or injury morphology scores at lower multiples of the free human $C_{max}$ compared to each of their non-toxic comparators, a finding that was repeated across 3 donors (Table 2). Furthermore, immunofluorescence microscopic imaging for markers of apoptotic cell death (caspase 3/7) and mitochondrial injury measured by visualizing reduction of tetramethylrhodamine methyl ester (TMRM) accumulation (Fig. 4) provided confirmation of toxicity and, in many cases,

**Table 2 Data for the matched pair analysis proposed by IQ MPS guidelines.**

| Drug | Albumin IC$_{50}$ | | | ALT | | | Morphology | | | IF imaging | | |
|---|---|---|---|---|---|---|---|---|---|---|---|---|
| | Donor 1 | Donor 2 | Donor 3 | Donor 1 | Donor 2 | Donor 3 | Donor 1 | Donor 2 | Donor 3 | Donor 1 | Donor 2 | Donor 3 |
| Clozapine | 33 | 91 | 67 | 300 | >300 | 100 | 100 | 100 | 100 | Apoptosis | Apoptosis | Apoptosis |
| Olanzapine | >300 | >300 | >300 | >300 | >300 | >300 | >300 | >300 | >300 | Apoptosis | No findings | No findings |
| Fialuridine | <0.1 | <1 | - | >300 | >300 | - | >300 | >300 | - | No findings | Steatosis | - |
| FIRU | >300 | >300 | - | >300 | >300 | - | >300 | >300 | - | No findings | Steatosis | - |
| Nefazodone | 140 | - | - | 300 | - | - | 300 | - | - | No findings | - | - |
| Buspirone | 282 | - | - | > 300 | - | - | >300 | - | - | Apoptosis | - | - |
| Sitaxsentan | 4.4 | 164 | >100 | 300 | 300 | 100 | 100 | 100 | 100 | Mitotoxicity | Mitotoxicity | Mitotoxicity |
| Ambrisentan | >300 | >1000 | >300 | >300 | >1000 | >300 | >300 | >1000 | >300 | No findings | No findings | No findings |
| Tolcapone | 3 | 39 | - | 100 | 30 | - | 10 | 100 | - | Mitotoxicity | Mitotoxicity | - |
| Entacapone | >300 | - | - | >300 | - | - | >300 | - | - | No findings | - | - |
| Troglitazone | 24 | 122 | - | 300 | 300 | - | 300 | 300 | - | Apoptosis | Apoptosis | - |
| Pioglitazone | 277 | >300 | - | >300 | >300 | - | >300 | >300 | - | Apoptosis | No findings | - |
| Trovafloxacin | 82 | 78 | 31 | 100 | 100 | 30 | 100 | 100 | 30 | Apoptosis | Apoptosis | Apoptosis |
| Levofloxacin | >15 | >45 | >45 | >15 | >45 | >45 | >15 | >45 | >45 | No findings | No findings | No findings |

Data from donors one, two, and three are presented in terms of multiples of the unbound human C$_{max}$ for each drug, to ease comparison within each pair. Included is the concentration causing a 50% reduction in albumin production, or the lowest concentration causing an increase in ALT protein or cellular morphology score. The dash indicates that the drug was not tested in donor 2 or 3. Apoptosis was defined as a concentration-dependent increase in caspase 3/7 staining, mitotoxicity was defined as a concentration-dependent reduction in TMRM staining and steatosis reflects an increase in Adipored staining. Representative images are contained in Supplementary Fig. 3.

provided some insight into the potential mechanism of toxicity. For example, the third-generation anti-infective trovafloxacin is believed to have an inflammatory component to its toxicity, potentially mediated by Kupffer cells, but this is only seen in animal models if an inflammatory stimulant such as lipopoly-saccharide (LPS) is co-administered[39]. Interestingly, immuno-fluorescence microscopic imaging of the Liver-Chip revealed that there was a concentration-dependent increase in caspase 3/7 staining following trovafloxacin treatment (Fig. 4); this supports a potential apoptotic component to its toxicity. Of note, levofloxacin, the lesser toxic structural analog, did not cause cellular apoptosis. The role of an activated immune system is considered to contribute to idiosyncratic DILI where a reactive metabolite forms an adduct that behaves like a hapten to activate the adaptive immune system[40] or directly activates innate immune cells (e.g., Kupffer cell) to increase inflammatory cytokine production such as TNFα[41]. To assess whether trovafloxacin was able to activate Kupffer cells in the absence of an inflammatory stimulant we measured IL-6 and TNFα in effluent from the bottom channel of Liver-Chips from each of the three donors. The measured levels in vehicle-treated Liver-Chips were low (300–400 pg/mL for IL-6 and 5–25 pg/mL for TNFα), indicative of non-activated cells and comparable to literature values[42]. We were also unable to see any concentration-dependent increase in cytokine production following treatment with either trovafloxacin or levofloxacin.

Together, these data support the Liver-Chip's value as a predictor of drug-induced toxicity in the human liver and demonstrate that this experimental system meets the basic IQ MPS criteria for preclinical model functionality. However, in addition to the seven matched pairs, the IQ MPS guidelines require that an effective human MPS DILI model can predict liver responses to six additional small-molecule drugs associated with clinical DILI. We only analyzed the effects of five of these drugs (diclofenac, asunaprevir, telithromycin, zileuton, and lomitapide) because the reported mechanism of toxicity of one of them (pemoline) is immune-mediated hypersensitivity[43], and this would potentially require a more complex configuration of the Liver-Chip containing additional immune cells. We also were unable to obtain one of the suggested drugs, mipomersen, from any commercial vendor; however, we tested lomitapide as an alternate, as both produce steatosis by altering triglyceride export, and lomitapide is known to induce elevated ALT levels[44].

Results obtained with these drugs are presented in Table 3, with toxicity values indicating the lowest concentration at which toxicity was detected. Lomitapide was highly toxic when tested over all included concentrations down to 0.1× human plasma C$_{max}$, with all Liver-Chips showing signs of toxicity following five days of dosing. While telithromycin displayed a decrease in albumin along with a concomitant increase in ALT and morphological injury score, diclofenac and asunaprevir induced concentration and time-dependent changes in albumin and injury scores, but no elevation of ALT was seen with these drugs. Hepatotoxicity was also confirmed with immunofluor-escence microscopy, which revealed apoptosis-mediated cell death following exposure to diclofenac, asunaprevir, or telithromycin. However, the Liver-Chip was unable to detect hepatotoxicity caused by zileuton, a treatment intended for asthma. The exact mechanism of toxicity of zileuton is unknown, but it likely involves production of intermediate reactive metabolites due to oxidative metabolism by the cytochrome P450 isoenzymes 1A2, 2C9, and 3A4[45]. Although zileuton is >93% plasma protein bound[46], we do not believe this was responsible for the lack of toxicological effect, as we were able to detect toxicities induced by other highly protein-bound drugs in the test set.

**Table 3 Results obtained with the expanded drug list.**

| Drug | Albumin IC$_{50}$ | | ALT | | Morphology | | IF imaging | |
|---|---|---|---|---|---|---|---|---|
| | Donor 1 | Donor 2 | Donor 1 | Donor 2 | Donor 1 | Donor 2 | Donor 1 | Donor 2 |
| Asunaprevir | 190 | - | >1000 | - | 1000 | - | Apoptosis | - |
| Benoxaprofen | 8 | 20 | >100 | 100 | <0.1 | 100 | Apoptosis | Apoptosis |
| Beta-Estradiol | >300 | - | >300 | - | >300 | - | No findings | - |
| Chlorpheniramine | >300 | - | >300 | - | >300 | - | No findings | - |
| Diclofenac | 384 | - | >1000 | - | 1000 | - | Apoptosis | - |
| Labetalol | 26 | 40 | >100 | 100 | >100 | 100 | Apoptosis | Apoptosis |
| Lomitapide | 4 | - | >1000 | - | <0.1 | - | No findings | - |
| Simvastatin | <0.1 | <10 | >300 | >300 | >300 | >300 | No findings | Apoptosis |
| Stavudine | >60 | 132 | 300 | >300 | 300 | 300 | No findings | Mitotoxicity |
| Tacrine | 76 | >25 | >300 | >1000 | >300 | >1000 | Apoptosis | Apoptosis |
| Telithromycin | 13 | - | 30 | - | 100 | - | Apoptosis | - |
| Ximelagatran | 32 | >300 | 30 | >300 | 100 | >300 | No findings | No findings |
| Zileuton | >100 | 210 | >100 | >300 | >100 | >300 | No findings | No findings |

Data are presented as the lowest unbound human C$_{max}$ multiplier causing a 50% reduction in albumin production, or the lowest concentration causing an increase in ALT release or morphology score. The dash indicates that the drug was not tested in donor 2. Apoptosis was defined as a concentration-dependent increase in caspase 3/7 staining and mitotoxicity was defined as a concentration-dependent reduction in TMRM staining. Representative images are contained in Supplementary Fig. 3.

**Improved sensitivity for DILI prediction compared to spheroids and animal models.** After fulfilling the major criteria of the IQ MPS affiliate guidelines, we considered the Liver-Chip to be qualified as a suitable tool to predict DILI during preclinical drug development. However, we wished to also quantify the performance of the Liver-Chip in the predictive toxicology context. To do so, we expanded the drug test to include eight additional drugs (benoxaprofen, beta-estradiol, chlorpheniramine, labetalol, simvastatin, stavudine, tacrine, and ximelagatran) that were found to induce liver toxicity clinically, despite having gone through standard preclinical toxicology packages involving animal models prior to first-in-human administration. Importantly, the toxicities of these 8 drugs have been shown to be poorly predicted by hepatic spheroids[31,32].

We proceeded to quantify any observed toxicity across the combined and blinded 27-drug set as a margin of safety (MOS)-like figure by taking the ratio of the minimum toxic concentration observed to the clinical C$_{max}$. We obtained the minimum toxic concentration by taking the lowest concentration identified by each of the primary endpoints—i.e., IC$_{50}$ values for the decrease in albumin production, the lowest concentration at which we observed an increase in ALT protein, and the lowest concentration at which we observed injury via morphology scoring (Table 4). Minimum toxic concentrations generally corresponded to day seven values, although day three values were occasionally lower. We then compared the MOS-like figures against a threshold value of 50 to categorize each compound as toxic or non-toxic, as previously reported for 3D hepatic spheroids, which used a similar threshold[31]. Analyzed in this manner, we found that, in addition to the drugs assessed as part of the IQ MPS-related analysis, the Liver-Chip correctly determined labetalol and

**Table 4 Calculation of margin of safety (MOS)-like figures.**

| Drug | Chip MOS donor 1 | Chip MOS donor 2 | Chip MOS both donors | Spheroid MOS |
|---|---|---|---|---|
| Ambrisentan | >3 | >10 | >10 | >127 |
| Asunaprevir | 4 | – | – | – |
| Benoxaprofen | 0.001 | 0.3 | 0.001 | >0.7 |
| Beta-Estradiol | >5 | – | – | 22,500 |
| Buspirone | >15 | – | – | 16,300 |
| Chlorpheniramine | >200 | – | – | 2141 |
| Clozapine | 1.8 | 5 | 1.8 | 14.5 |
| Diclofenac | 1.8 | – | – | 6.1 |
| Entacapone | >6 | – | – | 46.5 |
| Fialuridine | 0.1 | 2.8 | 0.1 | 12.3 |
| FIRU | >108 | >108 | >108 | – |
| Labetalol | 26 | 22 | 22 | 0.41 |
| Levofloxacin | >11.3 | >33.7 | >33.7 | >20 |
| Lomitapide | 0 | – | – | – |
| Nefazodone | 1.4 | – | – | 6.8 |
| Olanzapine | >20 | >2 | >20 | – |
| Pioglitazone | 2.8 | > 2.8 | 2.8 | > 5.3 |
| Simvastatin | 17 | 45 | 17 | 460 |
| Sitaxsentan | 0.04 | 1.6 | 0.04 | 8.7 |
| Stavudine | 247 | 107 | 107 | >144 |
| Tacrine | >12 | >12 | >12 | 696 |
| Telithromycin | 6 | – | – | – |
| Tolcapone | 0.004 | 0.05 | 0.004 | 0.3 |
| Troglitazone | 0.03 | 0.1 | 0.03 | 2.3 |
| Trovafloxacin | 20 | 19 | 19 | >24.9 |
| Ximelagatran | 30 | 300 | 30 | 335 |
| Zileuton | >7 | 15 | 15 | >7.7 |

The analysis was carried out using the free IC$_{50}$ concentration of the drug tested in the assay divided by the total concentration of drug in human plasma at C$_{max}$[31].

benoxaprofen to be hepatotoxic, a response that was consistent across donors one and two and was indicated primarily by a reduction in albumin production. However, we found that simvastatin and ximelagatran were only toxic in one of the donors tested, again showing the importance of including multiple donors during the risk assessment process. Overall, the Liver-Chip correctly predicted toxicity in 12 out of 15 toxic drugs that were tested using two donors, yielding a sensitivity of 80% on this drug set. This was almost double the sensitivity of 3D hepatic spheroids for the same drug set (42%) based on previously published data[31,47,48], a preclinical model that is currently widely used in pharma and was only able to correctly identify 8 out of the 19 toxic drugs in the set (Table 5). Importantly, the Liver-Chip also did not falsely mark any drugs as toxic (specificity of 100%), whereas the 3D hepatic spheroids did (only 67% specificity)[31]; such false positives can greatly limit the usefulness of a predictive screening technology because of the profound consequences of erroneously failing safe and effective compounds. Interestingly, the three drugs not detected by Liver-Chip —levofloxacin, stavudine, and tacrine—were not detected as toxic drugs in spheroids either, suggesting that the Liver-Chip may subsume the sensitivity of spheroids and that their toxicities could involve other cells or tissues not present in these models. It is important to note that each of the toxic drugs tested was historically evaluated using animal models, and in each case the considerations and thresholds were deemed relevant for that drug to have an acceptable therapeutic window and thus progress into clinical trials. The ability of the Liver-Chip to flag 80% of these drugs for their DILI risk at their clinical concentrations represents a remarkable improvement in model sensitivity that could drive better decision making in preclinical development.

We examined each of the toxic drugs that were missed by the Liver-Chips to identify opportunities for future improvement. Using the threshold of 50 for determining toxicity, which we chose to compare our results to those from a past hepatic spheroid study, led to stavudine being classified as a false negative. Tacrine is a reversible acetylcholinesterase inhibitor that undergoes glutathione conjugation by the phase II metabolizing enzyme glutathione S-transferase in liver. Polymorphisms in this enzyme can impact the amount of oxidative DNA damage, and the M1 and T1 genetic polymorphisms are associated with greater hepatotoxicity[49]. It is not known if either of the two hepatocyte donors used in this investigation have these polymorphisms, but

the Liver-Chip was able to detect increased caspase 3/7 staining— indicative of apoptosis at the highest tested concentrations— although these changes were not associated with any release of ALT or decline in albumin. Levofloxacin, a fluoroquinolone antibiotic, was proposed by the IQ MPS affiliate as a lesser hepatotoxin compared to its structural analog trovafloxacin, but it is classified as high clinical DILI concern in Garside's DILI severity category labeling[50]. Indeed, there are documented reports of hepatotoxicity with levofloxacin, but these occurred in individuals aged 65 years and above[51], and a post-market surveillance report documented the incidence of DILI to be less than 1 in a million people[52]. It is therefore reasonable to assume that the negative findings in both the Liver-Chip and spheroids may correctly represent clinical outcome.

**Accuracy improved by accounting for drug-protein binding.** When calculating the MOS-like values in the preceding section, we followed the published methods used for evaluating 3D hepatic spheroids[31], but these do not consider protein binding. Because the fundamental principles of drug action dictate that free (unbound) drug concentrations drive drug effects, we explored an alternative methodology for calculating the MOS-like values by accounting for protein binding using a previously reported approach[36]. Accordingly, we reanalyzed the findings for the 27 drugs in our study by accounting for protein binding. We compared the free fraction of drug concentration dosed in the Liver-Chip employing a medium containing 2% fetal bovine serum to the free fraction of the plasma $C_{max}$. By reanalyzing the Liver-Chip results using this approach and setting the threshold value to 375 (which we selected to maximize sensitivity while avoiding false positives), we obtained improved chip performance: a true positive rate (sensitivity) of 77 and 73% in donors one and two, respectively, and a true negative rate (specificity) of 100% in both donors (Table 6). Importantly, the sensitivity increased to 87% when including the 18 drugs tested in both donors, and this enabled detection of stavudine's toxicity. Applying the same analysis to spheroids and similarly selecting a threshold to maximize sensitivity while maintaining 100% specificity yielded a sensitivity of only 47%. Remarkably, the Spearman correlation between the two-donor Liver-Chip assay and the Garside DILI severity scale yielded a value of 0.78 when using the protein-binding-corrected analysis, whereas it was only 0.43 when

**Table 5 Sensitivity and specificity determination.**

| Model | True positive | True negative | False positive | False negative | Sensitivity (%) | Specificity (%) |
|---|---|---|---|---|---|---|
| Chip donor 1 | 16 | 5 | 0 | 6 | 73 [51.4–86.8%] | 100 |
| Chip donor 2 | 9 | 3 | 0 | 6 | 60 [35.3–80.3%] | 100 |
| Chip both donors | 12 | 3 | 0 | 3 | 80 [54.2–92.8%] | 100 |
| Spheroid | 8 | 2 | 1 | 11 | 42 [22.9–64.0%] | 67 |

Predictive performance for chips and spheroids as determined by published analysis[31], which does not account for protein binding, and setting the threshold on the MOS-like values at 50 for both chips and spheroids (a value above 50 would indicate a drug as negative for toxicity).

**Table 6 Sensitivity and specificity determination.**

| Model | True positive | True negative | False positive | False negative | Sensitivity (%) | Specificity (%) |
|---|---|---|---|---|---|---|
| Chip donor 1 | 17 | 5 | 0 | 5 | 77 [56.1–89.8%] | 100 |
| Chip donor 2 | 11 | 3 | 0 | 4 | 73 [47.5–89.0%] | 100 |
| Chip both donors | 13 | 3 | 0 | 2 | 87 [61.5–96.0%] | 100 |
| Spheroid | 9 | 3 | 0 | 10 | 47 [27.0–68.5%] | 100 |

Predictive performance as determined by considering free (unbound) drug concentrations and setting the threshold on the MOS-like values at 375 for chips and at 2250 for spheroids. 95% confidence intervals are shown for the sensitivity values.

using the lower threshold. Thus, the protein-binding-corrected approach not only produces higher sensitivity but also rank-orders the relative toxicity of drugs in a manner that corresponds better with the DILI severity observed in the clinic. This observation supports the validity of this analysis approach and its superiority over the uncorrected version. In short, these results provide further confidence that the Liver-Chip is a highly predictive DILI model and is superior in this capacity to other currently used approaches.

**The economic value of more predictive toxicity models in preclinical decision making.** In addition to increasing patient safety, better prediction of candidate drug toxicity can improve the economics of drug development by reducing clinical trial attrition and increasing pharma research and development (R&D) productivity. We sought to quantify the potential economic impact of the Liver-Chip resulting from its enhanced predictive validity by constructing an economic value model of drug development that captures decision quality during preclinical development (Fig. 2). We describe the structure of this model in the Methods section and provide an interactive form of the full model in the Supplementary Materials (Supplementary Data 2).

To estimate the economic impact of incorporating the Liver-Chip into preclinical research, we observed that DILI currently accounts for 13% of clinical trial failures that are due to safety concerns[24]. The present study revealed that the Liver-Chip, when used with two donors and analyzed with consideration for protein binding, provides a sensitivity of 87% when applied to compounds that evaded traditional safety workflows. Combining these figures suggests that adding the human Liver-Chip to existing workflows to test for DILI risk could lead to 11.3% fewer toxic drugs entering clinical trials. We modeled this improvement by correspondingly lowering the model's false negative rate (FNR) parameter that describes the toxicology testing that occurs between preclinical testing and Phase I clinical trials. We then computed the net present value (NPV) of the new simulated portfolio and compared it to the NPV of the base case to capture the increase in R&D productivity. Improvements in NPV result from an increase in the number of approved drugs, which is partially offset by the added cost of Organ-Chips experiments as well as higher clinical trial costs resulting from candidates progressing further in their clinical testing. This computation resulted in a predicted NPV uplift of 2.8% (1.9–3.1%, CI 95%) due to the incorporation of the Liver-Chip in DILI prediction (Supplementary Table 2 lists results for a broad range of FNR values).

We next estimated the potential impact of this value uplift on the broader small-molecule drug-development industry by applying it to the global Pharma investment in R&D. In 2021, global R&D investment was approximately $196 m per year[53] of which around 56%[54] was related to small-molecule drugs. With these assumptions, the model predicts that utilizing the Liver-Chip across all small-molecule drug-development programs for DILI prediction could generate the industry around $3 billion annually due to increased R&D productivity ($2.1B - $3.4B, CI 95%). Since the economic model relies on historical attrition rates and costs, we assessed the robustness of the above predictions with respect to the model's inputs by performing a mathematical sensitivity analysis. This analysis revealed that model outputs vary in a near-linear fashion across reasonable input parameter sets, which in turn causes the model's predictions to stay qualitatively consistent across a wide range of parameter choices. The details of this analysis and its results are included in the model (Supplementary Data 2).

The economic model also permits us to estimate the financial impact of Organ-Chip technology as the predictive validity of additional toxicology models is evaluated similarly to our work here on the Liver-Chip. We were particularly interested in the potential impact of four additional Organ-Chips that address the remaining top causes of safety failures—cardiovascular, neurological, immunological, and gastrointestinal toxicities, which together with DILI account for 80% of trial failures due to safety concerns[24]. If we assume similar sensitivity for these four additional models as we found for the Liver-Chip (87%), the model estimates that Organ-Chip technology could generate the industry over $24 billion annually through increased R&D productivity. These figures present a compelling economic incentive for the adoption of Organ-Chip technology alongside considerations of patient safety and the ethical concerns of animal testing.

## Discussion

Numerous authors have argued that Organ-Chip technology has the potential to substantially improve drug discovery and development[55], but although many major pharmaceutical companies have already invested in the technology, routine utilization is limited[56]. This may be due to several factors, including the absence of end-to-end investigations showing that Organ-Chips replicate human biological responses in a robust and repeatable manner; demonstrations that Organ-Chip performance exceeds that of existing preclinical models across a suitably broad set of compounds; and illustrations of ways to implement the technology within routine preclinical workflows. Furthermore, the broader stakeholder group—especially budget holders—need assurance that there will be an attractive return on investment and an increase in R&D productivity that may mitigate the pharmaceutical industry's widely documented productivity crisis[57–59]. This study aims to address these four concerns.

We particularly report here on the systematic evaluation of the validity of Organ-Chips for DILI prediction against criteria designed by a third party of experts. To our knowledge, no MPS has been evaluated against 27 small-molecule drugs in a single study involving three different human donors and hundreds of chips, making this study the largest reported evaluation of Organ-Chip performance. In this evaluation, the Liver-Chip has demonstrated that it can correctly distinguish toxic drugs from their non-toxic structural analogs, and, across a blinded set of 27 small molecules, it displayed a true positive rate of 87%, a specificity of 100%, and a Spearman correlation of 0.78 against the Garside DILI severity scale when two donors are used, and data are corrected for protein binding. Importantly, these data were independently verified by two external toxicologists. Said differently, the Liver-Chip detected nearly 7 out of every 8 drugs that proved hepatotoxic in clinical use despite having been deemed to have an appropriate therapeutic window by animal models; the Liver-Chip similarly detected 2 out of 4 such drugs that were additionally missed by 3D hepatic spheroids. We therefore believe that these findings advocate the routine use of the human Liver-Chip in drug discovery programs to enhance the probability of clinical success while improving patient safety. This would be achieved by more accurately categorizing risk associated with a candidate drug to provide valuable data to support a 'weight-of-evidence' argument both for entry into the clinic as well as for starting dose in Phase I. Such added evidence could potentially remove any safety factor applied because of a liver finding in an animal model[60,61]. In turn, this would reduce overall cost and time in the preclinical development process.

A unique feature of this work is the demonstration of the throughput capability of Organ-Chip technology using automated

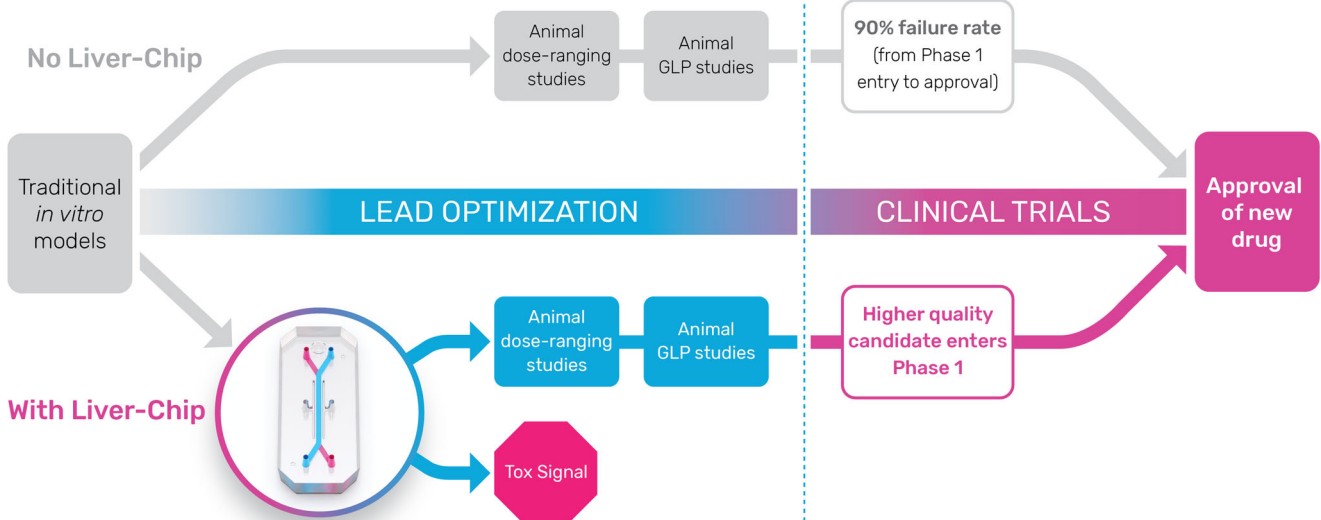

**Fig. 5 Proposed positioning of the Liver-Chip within a typical pharma preclinical workflow.** Typically, pharma utilizes a series of in vitro tests to guide chemical optimization ahead of animal testing. Promising drug candidates then progress to dose-range finding studies ahead of the required studies to enable regulatory approval to enter clinical trial. With the data presented in this investigation, Liver-Chip would be best placed in between the in vitro tests and dose-range finding animal studies. A drug candidate that did not show toxicity in the Liver-Chip, would increase confidence of the scientist that it can pass through animal testing without a liver toxicity flag and proceed into the clinic with a lower likelihood of clinical hepatic signals. A drug candidate that did show toxicity in the Liver-Chip would encourage scientists to stop and think about the relevance of the toxicity to the therapeutic indication and whether there was a potential margin between this finding and the exposure required for clinical efficacy. This would continue to increase the confidence that candidate drugs are entering the phase I clinical trial process with a greater likelihood of approval and may also reduce animal usage by not conducting dose-range finding or regulatory studies.

culture instruments, as a total of 870 chips were created and analyzed. In terms of establishing effective workflows, scientists were placed into three teams: the first team prepared the drug solutions and supplied them in a blinded manner to the second team. The second team seeded, maintained, and dosed the Liver-Chips while carrying out various morphological, biochemical, and genetic analyses at the end of the experiment. The third team collected the effluents and performed real-time analyses of albumin and ALT as well as terminal immunofluorescence imaging using an automated confocal microscope (Opera Phenix; Perkin Elmer). In this manner, we were able to analyze and report the hepatotoxic effects of 27 drugs in 870 Liver-Chips that used cells from three human donors in a period of 20 weeks.

Based on this experience, we believe that the Liver-Chip could be employed in the drug-development pipeline during the lead optimization phase where projects have identified three-to-five chemical compounds that have the potential to become the candidate drug (Fig. 5). If data emerge showing that a chemical compound produces a toxic signal in the Liver-Chip, this will indicate to toxicologists that there is a high (~87%) probability that the compound would similarly cause toxicity in humans. This, in turn, would enable scientists to deprioritize these compounds from early in vivo toxicology studies (such as the maximum tolerated dose/dose-range finding study) and, consequently, reduce animal usage and advance the "fail early, fail fast" strategy. Importantly, the absence of false positives strengthens the argument that the Liver-Chips should also be adopted within the early discovery phase, as stopping drug candidates that are falsely determined to be toxic by less-robust preclinical models could result in good therapeutics never reaching patients.

Despite these positive findings, it should be acknowledged that the current chip material (PDMS) used in the construction of the Liver-Chip may be problematic for a subset of small molecules that are prone to non-specific binding. Although this study demonstrates that the material binding issue does not in practice greatly reduce the predictive value of the Liver-Chip DILI model,

work is currently underway to develop chips using materials that have a lower binding potential. Until such a chip is available, we recommend users assess potential PDMS binding using an acellular chip and measuring drug in the effluent channel using LC/MS to enable adjustment of workflow if required. It should also be recognized that many pharmaceutical companies have diversified portfolios, with only 40–50% now being small molecules. Consequently, further investigation of the Liver-Chip performance against large molecules and biologic therapies should be carried out. Integration of resident and circulating immune cells should add even greater predictive capability.

Finally, predictive models that demonstrate concordance with clinical outcomes should provide scientists and corporate leadership with greater confidence in decision making at major investment milestones. Our economic analysis revealed that supplementing existing preclinical models with human Liver-Chips for the prediction of small-molecule DILI could lead to a substantial economic impact, with broad adoption of the technology having the potential to generate an estimated $3 billion annually across the industry due to improved R&D productivity. Moreover, the analysis illustrates that the productivity gain could potentially extend to an estimated $24 billion annually if four additional Organ-Chip models are used to address the most common toxicities that result in drug attrition, and the additional Organ-Chips demonstrate a similar level of performance to the Liver-Chip. Taken together, these results suggest that Organ-Chip technology has tremendous potential to benefit drug development, improve patient safety, and enhance pharmaceutical industry productivity and capital efficiency. This work also provides a starting point for other groups that hope to validate their MPS models for integration into commercial drug pipelines

### Data availability
The source data needed to reproduce the plots in Figs. 1 and 2 can be found in Supplementary Data 8 and Supplementary Data 3, 4, and 7, respectively. The calculated

# ARTICLE

statistics presented in Fig. 2 can be found in Supplementary Data 5 and 6. The RNA-sequencing data have been deposited in the National Center for Biotechnology Information Gene Expression Omnibus (GEO) under accession number GSE207339. In Supplementary Data 2, we provide additional information about: (i) Analytic derivation of base case, (ii) Attrition parameter details, (iii) Pipeline and financial model, (iv) DILI and tox sensitivity analyses, (v) NPV to annualized financial value, (vi) References, (vii) Supplemental details on the method, (viii) Source for Supplemental results figure, and (ix) Discussion of broad potential of improved predictive toxicology. Finally, a summary of each drug used in each cycle of the study is reflected in Supplementary Data 1.

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

## Acknowledgements

We would like to recognize the wider Emulate team for their valuable discussion and support and are grateful for Aaron VanDevender's critical evaluation and review of the manuscript. We also thank Jackson Wells for his detailed editorial review.

## Author contributions

L.E., D.E.I., D.L., D.V.M., and J.D.S. designed the study; A.A., S.A.B., J.T.C., C.V.C., A.R.H., J.J., S.J., K.K.M., M.E.Q., M.K., A.C.R., W.T., M.W., and J.V. contributed to drug preparation, execution of the study, downstream processing of images, and data collection; J.J., S.J., S.R.J., G.K., V.J.K., C.Y.L., C.L., J.S.R., D.R.T., M.W., and J.F.K.S. chip seeding, maintenance, drug administration, and chip sampling; C.V.C., S.J., G.K., and V.J.K. provided leadership for experimental cycles; L.E., C.L., D.V.M., S.J., M.M.K., and J.D.S. contributed to the data analysis; A.R.H., L.E., D.E.I., C.Y.L., K.K.M., and M.W. contributed to writing the manuscript; D.L. and J.W.S. built the economic model and interpreted the Liver-Chip outcome on R&D productivity; K.-J.J., O.I., and P.K.M. performed a blinded data review and critically reviewed the manuscript.

## Competing interests

The authors declare the following competing interests: L.E., D.L., D.V.M., J.D.S., A.A., S.A.B., J.T.C., C.V.C., A.R.H., J.J., S.J., S.R.J., J.F.K.S., M.M.K., M.K., K.K.M., M.E.Q., A.C.R., W.T.S., M.W., G.K., V.J.K., C.Y.L., C. L., J.S.R., D.R.T., J.V., and K.-J.J. are employees or former employees of Emulate Inc. and may hold equity; D.E.I. is a founder, board member, SAB chair, and equity holder in Emulate Inc. J.W.S. is a shareholder and director of JW Scannell Analytics LTD and received payment from Emulate Inc. for contributing to this work. O.I. and P.K.M. have no competing interests to share.
