## [Peer Review File · Communications Medicine]

Reviewers' comments:

Reviewer #1 (Remarks to the Author):

The present manuscript set out to assess the applicability of microphysiological liver models for predictive toxicology using standard industrial guidelines. This approach is highly relevant for the organ on a chip field and constitutes a first important step towards biochip standardization and assay harmonization. The major claims of the study are that the implementation of Emulate's liver microphysiological systems meets the qualification guidelines and significantly reduces cost and increases R&D productivity. Both claims seem valid considering the small sample size (two donors) and the application of only a single organ-on-a-chip platform (not sure if other OoC systems are better or worse). The conclusions are convincing and the presented methods and results are appropriate. The presented results are novel and of great interest to the scientific and industrial communities.

Despite the importance of the conducted research some major questions still remain including:

A) Considering the importance of the presented study it is not well argued why only two donors have been used – at least three donors are the minimum sample size

B) at times the manuscript sounds like an industrial report where economic considerations are mainly presented. A more scientific communications are needed.

C) The albumin data need clarification! Albumin levels decrease over time and at day 7 are below recommended levels, thus indicating malfunctions.

D) Figure 1 – the standard variation seems quite low considering the high variances of the data distribution.

E) The urea data display sig. differences between donor 1 and 2 making a strong case for including an addition donor.

F) Liver specific characteristics are only analyzed at day 3 and 5 of culture – please elaborate why day 7 was not selected.

G) Remove the overstatement that liver chips enable multiple longitudinal measures since only 7 days of culture was investigated.

H) Please rephrase: page 7 sentence 3 – it is not clear why albumin production was used as a general measure of hepatocyte viability, provided that a linear decrease in albumin production was observed only on the chip platform.

I) Table 2 is good summary of the conducted body of work, however it is important to also provide access to the actual results in the suppl information. The statement e.g. “apoptosis” is simply not enough. Please provide scientific evidence!! Also clarify what empty cells, indicated by a dash, within the table mean?

J) Methods: Some experiments are conducted only in duplicates, which does not reflect Nature standards. Perhaps less cycles and more replicates would be better.

K) Also the main aspect of the work includes study design and data interpretation in regards to economic value analysis seems to be conducted by an analyst who is not included in the author list. Considering the focus of the manuscript regarding its economic impact this seems inappropriate.

The current work will have a big influence in the field and should be published following major revisions indicated above.

Reviewer #2 (Remarks to the Author):

The study by Ewart et al. truly is a “tour-de-force” to demonstrate what level of effort is required to “qualify” a liver microphysiological system (MPS) for predictive toxicology. From a regulatory standpoint, qualification is a rigorous standard, hence the scope of this study using over 700 liver MPS.

For my critiques, the first thing that struck me was that these standards are never clearly stated by the authors, only referred to vaguely with respect to the IQ-MPS. The abstract itself should clearly call out what the authors are using to meet their qualification standards e.g., physiologically relevant dose responses in toxicity as measured by X-fold change in biomarkers (i.e., decreased albumin and increased ALT), etc. Such parameters will be needed to define what is an acceptable level of variability, given the use of primary human hepatocytes

My second major critique is that only two hepatocyte donors (both male and Caucasian) were used. I recognize the challenges with this if one is to screen a large panel of compounds at multiple concentrations. With this in mind, I would like to see a deeper level of characterization at a genetic level and functional level. For example, CYP2D6 is highly polymorphic as are CYP2C9/2C19 along with UGTs and GSTs (which the authors do mention). At a minimum, pharmacogene genotyping should be conducted. For function, while the authors do perform gene expression profiling, protein activities are what will determine drug-induced toxicity in many cases. There are numerous assays one can apply here such as CYP cocktail LC-MS/MS (A Sensitive and Specific CYP Cocktail Assay for the Simultaneous Assessment of Human Cytochrome P450 Activities in Primary Cultures of Human Hepatocytes using LC-MS/MS (nih.gov)).

Minor comments.

1. Since many readers may not be familiar with the Emulate liver MPS, a better quality and clearly-labeled version of Figure 1A should be provided.
2. Related to major critique 2 and Figure 1E, how do these gene expression levels compare to published RNAseq on primary human hepatocyte in sandwich culture versus previous work by the authors or other investigators studying liver MPS (e.g., tissue chips groups at Pitt or Columbia)?
3. In the first mention of Table 1, the authors refer to 7 drugs identified by IQ-MPS-for clarity, these should be identified in Table 1 (e.g., highlighted). In addition, these 7 drugs were said to be tested at 8 different concentration (Supplemental Table 2) but some were tested at 5 or 6 concentrations.
4. Regarding toxicity evaluation by hepatocyte morphology, no cell images were presented. What does score of 4 versus 2 versus zero look like?
5. It was nice to see that trovafloxacin could induce hepatocyte injury without the addition of LPS but it would be even more powerful to see if this was associated with increased cytokines such as IL-6 and TNF α and could address the question of whether direct Kupffer cell-hepatocyte interactions are needed, since the Emulate MPS separates these cell populations.
6. The discussion about liver MPS impact on cost savings in drug development is appreciated but the explanation is rather complicated. The reader would benefit from a figure or flow chart diagramming how this technology can exert cost savings in the preclinical phases as well as potentially nonclinical phases to apply iPSC-based technology for patient-specific DILI. Included in this, there should be calculation of what MPS studies cost in comparison to standard approaches such as two species of GLP-tox.

Reviewer #3 (Remarks to the Author):

The paper of Ewart et al. evaluates the predictive value of Liver-Chips. The authors assessed the specificity and sensitivity of 780 Liver-Chips to predict drug-induced liver injury (e.g. toxicity). Next to this pharmacologic evaluation, the authors also added a computational economic value analysis that suggest that Liver-Chip could generate \$3 billion annually for the pharmaceutical industry. Moreover, the authors claim that Organ-Chip technology could create an additional productivity value of \$24 billion. My review specifically focuses on this economic evaluation. While I general agree with the approach of taking an existing financial model and replacing relevant parameters with the results of the pharmacologic analysis, critical information are missing to understand the economic analysis.

1) Underlying framework: The authors describe that they use a pre-economic value model that was developed by Dr Jack Scannell without giving more information on the model (nor a source, if I see correctly). This makes it difficult to understand the baseline framework. Is the Dr Jack Scannell model peer reviewed material/published (Source 4)? An overview figure of the baseline model may help and should include the framework as well as baseline parameters. (Line 317, is source 45 correct?)

2) Input parameters: The authors say: “we then parameterized the model using the results of the present study” (line 317). Please specify the exact parameters as they are likely to have a strong impact on the final estimate of \$3 billion.

3) Sensitivity analysis: I would expect a sensitivity analysis as part of an thorough economic analysis.

4) Methods section: The methods section only describes the pharmacologic evaluation. I suggest adding a section for the economic analysis.

5) Discussion/Estimated productivity gains: It would help to put the savings/added productivity of \$3 billion - \$24 billion into context.

a. I would expect a range, rather than an absolute number. As such, the number is presented with a certainty that does not reflect the work that has been conducted.

b. The authors say that “the model estimates that Organ-Chip technology could generate the industry over \$24 billion annually” - This is a bold claim considering the rough assumptions that have been applied.

c. As far that I can see, the study’s estimates exclusively base on a change in drug attrition. Are also changes in other parameters expected because of the technology? Will the direct cost (such as material etc.) increase because of Liver-Chip?

d. How do the findings relate to the cited eroom’s law? E.g. why do the authors expect that Liver-chip will increase productivity rather than add to cost?

RESPONSE TO REVIEWERS

Reviewers' comments:

Reviewer #1 (Remarks to the Author):

The present manuscript set out to assess the applicability of microphysiological liver models for predictive toxicology using standard industrial guidelines. This approach is highly relevant for the organ on a chip field and constitutes a first important step towards biochip standardization and assay harmonization. The major claims of the study are that the implementation of Emulate's liver microphysiological systems meets the qualification guidelines and significantly reduces cost and increases R&D productivity. Both claims seem valid considering the small sample size (two donors) and the application of only a single organ-on-a-chip platform (not sure if other OoC systems are better or worse). The conclusions are convincing, and the presented methods and results are appropriate. The presented results are novel and of great interest to the scientific and industrial communities.

Despite the importance of the conducted research some major questions still remain including:

A) Considering the importance of the presented study it is not well argued why only two donors have been used – at least three donors are the minimum sample size

One of the questions that we wished to answer in our study's design is how many donors are necessary for the technology to be used practically in Pharma workflows. Accordingly, we sought to evaluate performance of the Liver-Chip with a single donor, two donors, and if necessary, we planned to explore additional donors. We saw a clear improvement in the model's performance when we used two donors compared to only a single donor. However, the level of predictive performance that we obtained using two donors (87% sensitivity, 100% specificity) seemed very adequate for real-world use, and so we stopped exploring this dimension any further. That said, in response to your request, we have now carried studies with an additional donor for three matched pairs of drugs (Levofloxacin-Trovafloxacin, Clozapine-Olanzapine, Ambrisentan-Sitaxsentan) to explore what further benefit could be obtained with a third donor. Unlike donor one and two, the third donor was female, of a different ethnicity and was also older (details of all three donors are shown in Supplementary Table S1). Importantly, the data analysis results were consistent with the previously tested donors and did not add any significant statistical value beyond the two-donor study. These findings underscore the reliability of our model and supports the need for at least two different donors for this type of safety assessment. These findings are now described in Methods, Results, and Discussion.

B) At times the manuscript sounds like an industrial report where economic considerations are mainly presented. A more scientific communications are needed.

The tone of the article and its scientific integrity are very important to us, and we appreciate your thoughts on this. While it may indeed be unusual for a biological study to discuss economics, we purposefully included the economic analysis because we deemed it to be an integral aspect of the message of this article. In particular, the Organ-on-Chip field now thankfully has many published demonstrations that highlight case studies where Chips solve

important biological problems. However, the potential users of these technologies in Pharma seem unclear whether these are cherry-picked cases, or whether the technology ought to be incorporated into standard workflows; that is why we were driven to evaluate predictive performance quantitatively as well. However, even with this quantification (87% sensitivity, etc.), we found that the overall message wasn't coming through – it wasn't clear to our co-authors and others who we ran the draft article by whether the 87% mattered at all, especially given that DILI is only one of several reasons for clinical trial safety failures. This is why we constructed the economic model and seek to use these data to speak to different stakeholders within Pharma. The purpose of listing numbers such as the \$3B annual increase due to R&D productivity was to illustrate that the biological results are meaningful to the drug development process, even though we only looked into one very specific aspect of the development process, the prediction of drug-induced liver injury. Importantly, one of the Reviewers felt this analysis was indeed valuable but requested additional information relating to the methodology of the economic modeling, which we now include in the Methods and Discussion.

C) The albumin data need clarification! Albumin levels decrease over time and at day 7 are below recommended levels, thus indicating malfunctions.

First, it is important to note that a broad review of published albumin data from other microphysiological systems, which can include Organ-Chips, illustrates that albumin data are often displayed as a percentage change from vehicle control. As such, these data do not allow the reader to ascertain the physiological level of albumin production. Our data are presented in accordance with the IQ MPS affiliate guidelines, that is reporting the absolute albumin concentration across the duration of our study. Second, our designation at day 7 relates to 7 days post-vehicle administration whereas this equates to day 12 of culture for hepatocytes. While we observed that one donor maintained albumin levels above the computed target level of 20 µg per million hepatocytes per day for this full 7-day treatment period, one of the other original two donors and a third new donor maintained physiologically relevant levels at earlier times but reduced below this target level by day 7 of drug or vehicle administration (day 12 in culture). Based on our validation requirements, this indicates that the hepatocytes from these donors are coming to the end of their productive life, and we would not use data from these donors beyond 12 days in culture. Moreover, we believe that measurement and reporting of actual albumin levels should be a quality control criterion for each donor such that those donors that fail to reach the minimum level of albumin production at the start of the drug (or vehicle treatment) should not be included in an experiment.

D) Figure 1 – the standard variation seems quite low considering the high variances of the data distribution.

Thank you for bringing this to our attention and we apologize for this typographical error and in the revised manuscript we have carefully re-checked and corrected any typos including the corresponding text that refers to the error bars of the corresponding plots.

As we describe in the statistical analysis section of the Methods, the error bars shown in Figure 1d, 1e and 2a correspond to the Standard Error of the Mean (SEM). The SEM is defined as: $SE = \frac{SD}{\sqrt{n}}$ where SD is Standard Deviation and n is the number of samples. The division of the SD with the square root of the number of samples, explains the small size of the error bars in the corresponding bar plots.

E) The urea data display sig. differences between donor 1 and 2 making a strong case for including an addition donor.

As suggested, we have performed a further experimental cycle that uses a third human hepatocyte donor. We have measured urea in the top channel effluent of the vehicle chips from this donor on days 1, 3 and 7 and have updated Figure 2e as well as highlighted areas of the text as appropriate. Urea levels in donor 3 were consistent across days 1, 3 and 7 and were towards the upper end of the computed physiological range. Donor 3 produced less urea on day 1 compared to donor 1 and 2 but levels were more comparable on days 3 and 7.

F) Liver specific characteristics are only analyzed at day 3 and 5 of culture – please elaborate why day 7 was not selected.

As described above, we used production of physiological levels of albumin production ($\geq 20 \mu\text{g}$ albumin per million hepatocytes per day) as a validation criterion in our study and this dropped in some donors by day 7. We therefore originally analyzed RNA at day 3 and 5 post-vehicle administration because we wanted to be sure that the hepatocytes would be viable and obtain sufficient RNA for analysis. As discussed above and throughout the manuscript, we appreciate that some of our chips can indeed go to Day 7 post-vehicle or drug administration and still meet the validation criteria. Therefore, we have performed an additional experiment with this donor, taking it out to day 7 post-vehicle administration and we now provide these RNA data in revised **Figure 2f**. Additionally, we performed RNAseq for donor 3 also on day 3 and 7. Data on these graphs have been replotted to express the data as fold-change relative to the freshly thawed hepatocytes, which was an additional step we undertook to provide further interpretation of the data which we cover in the revised Results section.

G) Remove the overstatement that liver chips enable multiple longitudinal measures since only 7 days of culture was investigated.

We are grateful to the reviewer for highlighting this statement and we recognize that it may be unclear. In response to this concern, we have removed the word “longitudinal” and replaced it with “on days 1, 3 and 7 post-vehicle or drug administration” because effluent samples were collected, and albumin and ALT were measured on each of these days.

H) Please rephrase: page 7 sentence 3 – it is not clear why albumin production was used as a general measure of hepatocyte viability, provided that a linear decrease in albumin production was observed only on the chip platform.

Protein synthesis, with albumin being one of the major proteins, is an important function of the liver. Because secreted albumin can be measured easily in cell culture medium, it is often used as a measure of hepatocellular functionality in the liver field. Albumin production is a multi-step process involving transcription, translation, and rough endoplasmic reticulum processing. Thus, if albumin production is inhibited, it can indirectly indicate reduced cellular viability. We welcome your attention on the language used on page 7, sentence 3 and have replaced “viability” with “functionality”.

I) Table 2 is good summary of the conducted body of work, however it is important to also provide access to the actual results in the suppl information. The statement e.g.

“apoptosis” is simply not enough. Please provide scientific evidence!! Also clarify what empty cells, indicated by a dash, within the table mean?

As requested, we have updated the legend for **Table 2** and **Table 3** to clarify that a dash represents that the drug was not tested in that donor. In addition, we have added clarifying text describing the results regarding the apoptosis or mitotoxicity labels and have also added an additional **Supplementary Figure S3** which shows representative immunofluorescent images from a low, medium and high concentration of each drug in each of the donors the drug was tested on.

J) Methods: Some experiments are conducted only in duplicates, which does not reflect Nature standards. Perhaps less cycles and more replicates would be better.

A key question we were trying to address with this study design was the concentration of a drug that would drive a hepatotoxic response and therefore, we wanted to assess within the technical constraints of our study the largest number of drug concentrations that could be tested. We generated 3 different “dose-response” synthetic data sets, performed a curve fit and calculated the Root Mean Square Error. Our analysis demonstrated that 8 concentrations in two replicates generated the optimal outcome, as described within the Experimental Setup section of the Methods. However, in response to this Reviewer's concern, we have performed an additional experimental cycle with a third hepatocyte donor. Based on learnings regarding the concentration-response relationships obtained from donors one and two, we designed this experiment with only 4 concentrations but included 3 replicates at each concentration. We show that the drug responses have comparable effects to those measured in chips created with cells from donors one and two. The data now from 3 independent donors are included in **Table 2**.

K) Also the main aspect of the work includes study design and data interpretation in regards to economic value analysis seems to be conducted by an analyst who is not included in the author list. Considering the focus of the manuscript regarding its economic impact this seems inappropriate.

We have taken this comment to heart and have significantly expanded the description of the economic value model in the manuscript given the importance of this aspect of our analysis for the Pharma industry, and we have added Dr. Jack Scannell who is an expert economist with deep experience in evaluation of drug development processes to the author list. Moreover, we have now included a full Excel version of the model in the Supplementary Materials.

Final comment from reviewer 1: The current work will have a big influence in the field and should be published following major revisions indicated above.

Reviewer #2 (Remarks to the Author):

The study by Ewart et al. truly is a “tour-de-force” to demonstrate what level of effort is required to “qualify” a liver microphysiological system (MPS) for predictive toxicology. From a regulatory standpoint, qualification is a rigorous standard, hence the scope of this study using over 700 liver MPS.

For my critiques, the first thing that struck me was that these standards are never clearly stated by the authors, only referred to vaguely with respect to the IQ-MPS. The abstract itself should clearly call out what the authors are using to meet their qualification standards e.g., physiologically relevant dose responses in toxicity as measured by X-fold change in biomarkers (i.e., decreased albumin and increased ALT), etc. Such parameters will be needed to define

what is an acceptable level of variability, given the use of primary human hepatocytes. My second major critique is that only two hepatocyte donors (both male and Caucasian) were used. I recognize the challenges with this if one is to screen a large panel of compounds at multiple concentrations. With this in mind, I would like to see a deeper level of characterization at a genetic level and functional level. For example, CYP2D6 is highly polymorphic as are CYP2C9/2C19 along with UGTs and GSTs (which the authors do mention). At a minimum, pharmacogene genotyping should be conducted. For function, while the authors do perform gene expression profiling, protein activities are what will determine drug-induced toxicity in many cases. There are numerous assays one can apply here such as CYP cocktail LC-MS/MS (A Sensitive and Specific CYP Cocktail Assay for the Simultaneous Assessment of Human Cytochrome P450 Activities in Primary Cultures of Human Hepatocytes using LC-MS/MS (nih.gov)).

With the restriction on abstract word count, we have not included the qualification standards in the abstract, but these are referred to extensively in the main body of the manuscript, especially within the results section. We recognize that both donors used in our original submission were Caucasian males. In this revision, we have included a third donor who is female with an African American ethnicity. This donor performed similarly to donors one and two in the detection of drug-induced toxicity and therefore we have not conducted pharmacogene genotyping. We have however extended the RNAseq analysis to include donor two (male, Caucasian) and donor three (female, African American) comparing profile freshly thawed and out to day 7 post-drug or vehicle administration. We have also uploaded all of the RNASeq data from these two donors onto the GEO database (accession number GSE207339).

Minor comments.

1. Since many readers may not be familiar with the Emulate liver MPS, a better quality and clearly-labeled version of Figure 1A should be provided.

As requested, we now include a larger image of the chip as **Figure 1**. The other items that were previously in **Figure 1** have now become **Figure 2**. Subsequent figures have been relabeled accordingly.

2. Related to major critique 2 and Figure 1E, how do these gene expression levels compare to published RNAseq on primary human hepatocyte in sandwich culture versus previous work by the authors or other investigators studying liver MPS (e.g., tissue chips groups at Pitt or Columbia)?

We have increased the RNAseq analysis for donor two by generating data from freshly thawed hepatocytes. We have also included RNAseq analysis for donor three. We have also included in the Results an interpretation of these data relative to what we were able to identify in the literature. The Tissue Chips group at UPitt have only published gene expression data in their LAMPS liver disease model and they have not conducted RNA-seq in a small molecule toxicology study. We therefore have not referenced their work in this manuscript.

3. In the first mention of Table 1, the authors refer to 7 drugs identified by IQ-MPS-for clarity, these should be identified in Table 1 (e.g., highlighted). In addition, these 7 drugs were said to be tested at 8 different concentration (Supplemental Table 2) but some were tested at 5 or 6 concentrations.

We have highlighted the 7 pairs, 14 drugs, in bold and converted the text to italic to highlight the drugs from IQ MPS consortium and adjusted the text in the Results section to add clarity for the reader. Our study was designed to test 8 concentrations of each drug, although this was subject to solubility limits because we wanted to ensure the DMSO concentration was kept to 0.1%. We now explain more clearly in both Results and Figure/Table legends that studies involving 5 drugs (levofloxacin, tolcapone, benoxaprofen, labetaolol and telithromycin) involved testing of less than 8 concentrations because of their solubility limits. We have further clarified the concentrations tested in each cycle of the experiment in **Supplementary Table S2**.

4. Regarding toxicity evaluation by hepatocyte morphology, no cell images were presented. What does score of 4 versus 2 versus zero look like?

Thank you for highlighting this and we agree that this would be helpful to the reader. Thus, we have added representative images corresponding to the different scores in **Supplementary Figure 2**, in addition to providing our scoring matrix.

5. It was nice to see that trovafloxacin could induce hepatocyte injury without the addition of LPS but it would be even more powerful to see if this was associated with increased cytokines such as IL-6 and TNF α and could address the question of whether direct Kupffer cell-hepatocyte interactions are needed, since the Emulate MPS separates these cell populations.

This is an excellent question and to address it we analyzed top and bottom channel effluents from chips for IL-6 or TNF alpha in cycles 3, 4 and 5 that received either trovafloxacin or levofloxacin. We found that there was no concentration-dependent change in cytokine levels compared to vehicle in either trovafloxacin or levofloxacin treated chips across each donor tested. This would indicate that the hepatotoxic response seen in our study with trovafloxacin did not involve Kupffer cell activation. We now include these data in the Results and describe the effluent analysis approach in the Methods.

6. The discussion about liver MPS impact on cost savings in drug development is appreciated but the explanation is rather complicated. The reader would benefit from a figure or flow chart diagramming how this technology can exert cost savings in the preclinical phases as well as potentially nonclinical phases to apply iPSC-based technology for patient-specific DILI. Included in this, there should be calculation of what MPS studies cost in comparison to standard approaches such as two species of GLP-tox.

We have taken this comment seriously, and we have significantly expanded the description of the economic model in the manuscript. Moreover, we now include a full Excel version of the model in the Supplementary Materials together with instructions on how to use it. This “active” version of the model can indeed be used to capture the economics of any MPS safety assay (and not just the Liver-Chip). Thus, our hope is that the model that we are now sharing could broadly serve the MPS community to answer the questions that the Reviewer raised (e.g., with respect to iPSC-based technology). We are thankful for this comment, and we are excited to enable the community to conduct this sort of analysis and discussion.

Reviewer #3 (Remarks to the Author):

The paper of Ewart et al. evaluates the predictive value of Liver-Chips. The authors assessed the specificity and sensitivity of 780 Liver-Chips to predict drug-induced liver injury (e.g. toxicity). Next to this pharmacologic evaluation, the authors also added a computational economic value analysis that suggest that Liver-Chip could generate \$3 billion annually for the pharmaceutical industry. Moreover, the authors claim that Organ-Chip technology could create an additional productivity value of \$24 billion. My review specifically focuses on this economic evaluation. While I general agree with the approach of taking an existing financial model and replacing relevant parameters with the results of the pharmacologic analysis, critical information are missing to understand the economic analysis.

1) Underlying framework: The authors describe that they use a pre-economic value model that was developed by Dr Jack Scannell without giving more information on the model (nor a source, if I see correctly). This makes it difficult to understand the baseline framework. Is the Dr Jack Scannell model peer reviewed material/published (Source 4)? An overview figure of the baseline model may help and should include the framework as well as baseline parameters.(Line 317, is source 45 correct?)

This is fair concern and to address it we have expanded our discussion of the economic model developed by Dr. Jack Scannell and now include him as a co-author. Specifically, we significantly broadened the description of the model's construction and parametrization, added a diagram that illustrates the model's framework, and now include a full Excel version of the model in the Supplementary Materials so that readers can explore the details in full, as well as apply the model to analyze other MPS systems.

2) Input parameters: The authors say: "we then parameterized the model using the results of the present study" (line 317). Please specify the exact parameters as they are likely to have a strong impact on the final estimate of \$3 billion.

We have taken this question to heart, and the revised version of the manuscript provides full transparency and references for the different parameter choices. Key parameters and their sources are provided in the body of the manuscript, while the remainder are provided in the Excel file.

3) Sensitivity analysis: I would expect a sensitivity analysis as part of a thorough economic analysis.

We agree, and we have now added a sensitivity analysis with respect to key model parameters as a tab in the model's Excel version. This tab also allows the reader to adjust the sensitivity analysis (e.g., in terms of parameter ranges) to explore model sensitivity further. In addition, we now reference the sensitivity analysis in the body of the manuscript.

4) Methods section: The methods section only describes the pharmacologic evaluation. I suggest adding a section for the economic analysis.

We agree. The revised manuscript now includes a description of the economic model's construction, parameterization and application in the Methods section.

5a) Discussion/Estimated productivity gains: It would help to put the savings/added productivity of \$3 billion - \$24 billion into context. I would expect a range, rather than an

absolute number. As such, the number is presented with a certainty that does not reflect the work that has been conducted.

This is a fair concern. In response, we now report ranges for the economic predictions, which correspond to the confidence intervals that emerged from the biological analysis. For example, the \$3B figure is now associated with the range \$2B - \$3.2B (CI 95%). We were happy to see that this expanded analysis resulted in ranges that retain the qualitative implications of the original point estimates, further strengthening the analysis. We are grateful to the Reviewer for encouraging us to do this.

5b) The authors say that “the model estimates that Organ-Chip technology could generate the industry over \$24 billion annually”- This is a bold claim considering the rough assumptions that have been applied.

We agree that this there is a significant assumption contained in this estimate, particularly that the sensitivity of the four additional Organ-Chip models would be similar to the sensitivity that we observed for the Liver-Chip. We nevertheless think that it is helpful to the reader to see this sort of estimate – be it rough – because it highlights the importance of the MPS field as whole. We feel that this is a powerful message to end the paper on - especially given that we are not developing all 5 models– because we hope that the economic analysis can embolden our whole community including both model developers and potential users.

5c) As far that I can see, the study’s estimates exclusively based on a change in drug attrition. Are also changes in other parameters expected because of the technology? Will the direct cost (such as material etc.) increase because of Liver-Chip?

In terms of which model parameters would be affected by incorporating the Liver-Chip, we focused on the “first order” effect of reducing the false negative rate in preclinical testing (as we now detail in the manuscript). However, we expect that there will be some secondary effects. For example, we expect that some clinical failures due to lack of efficacy are due to drug developers being unable to increase dosage sufficiently to see a response due to safety concerns; accordingly, we expect some drop in failures due to efficacy, which isn’t currently captured. In other words, we have reason to believe that we are underestimating the potential impact of improved preclinical testing. However, we were comfortable with this because the impact already seems very significant, supporting our message to the community. Nevertheless, we now discuss these caveats. Also, regarding the cost of applying Organ-Chip technology: we now include an estimate of the full cost of Organ-Chip technology in the model, as this question indeed comes up often. Importantly, it turns out that the cost of using the technology pales in comparison to the potential upside.

5d) How do the findings relate to the cited eroom’s law? E.g. why do the authors expect that Liver-chip will increase productivity rather than add to cost?

We would be reluctant to relate the results in this paper too closely to Eroom’s Law (which was first published by our co-author Jack Scannell), particularly because Eroom’s law speaks to the economic trend over time and across many competing factors, including challenges in identifying new targets, depletion of “druggable” targets, etc. (e.g., arguments about the “low hanging fruit” having been picked). However, in specific regard to the cost of Liver-Chips versus an increase in productivity: the economic analysis clearly suggests that the increase in R&D productivity significantly outweighs the cost of incorporating Liver-Chip as part of the drug development pipeline. One must deviate from the biological study’s outcomes and/or

the model's parameterization very significantly in order to leave this "net positive" regime, and we now clearly make this point in the Discussion. We also hope that the Excel version of the economic model that we have now included will highlight this to readers and permit them to explore the analysis further. We thank the Reviewer for encouraging us to include this transparency and level of detail.

Reviewers' comments:

Reviewer #2 (Remarks to the Author):

I would like to thank the authors for their extensive revisions and expanding their studies to include a third donor. I am now comfortable with the responses to the questions of economic impact as well as nearly all my other comments/critiques. Two points that I would like to raise are: 1) The authors did not respond to my request for protein activity profiling, particularly CYP activities; 2) The authors did respond to my question about trovafloxacin-induced injury and cytokine generation by measuring both IL-6 & TNF-alpha. However, they noted no change and say that "...an additional inflammatory stimulus may be required..." My points to raise here are that the cytokine data are not shown-are they already elevated, suggesting Kupffer cell activation basally? Also, what is the "additional inflammatory stimulus"? Have the authors considered a larger cytokine panel or perhaps soluble CD163 or soluble mannose receptor?

Regarding the main points of Reviewer 1's comments, there is a great deal of congruence with my comments (e.g., use of only two donors, emphasis on economics, choice of DILI readouts and variability in data). In the revised manuscript and authors' responses, I feel the points raised have been adequately addressed.

Reviewer #3 (Remarks to the Author):

The manuscript has improved significantly and I appreciate the transparency of the economic analysis.

The economic section of the discussion could benefit from a couple of reflections:

- Uncertainty: Despite of including now confidence intervals and a sensitivity analysis, there is still a slight overselling of the productivity gain, which is somewhat unacademic. One important aspect is also economic and regulatory readiness that may effect realization of economic potential
- Cost: Are there any estimates of the investments needed? Will there be an increase in direct cost?
- Capacity: Will the capacity limits of the R&D pipeline naturally constrain the productivity gain? Especially for rare cancers, recruiting relevant patients for clinical trials is a massive constrain.

RESPONSE TO REVIEWERS (2nd Review)

Reviewer #2 (Remarks to the Author):

I would like to thank the authors for their extensive revisions and expanding their studies to include a third donor. I am now comfortable with the responses to the questions of economic impact as well as nearly all my other comments/critiques. Two points that I would like to raise are:

1) *The authors did not respond to my request for protein activity profiling, particularly CYP activities*

As we explain in the body of the manuscript, we have previously carried out CYP protein activity characterization of the human Liver-Chips used in the present study, and this information has already been published in two separate peer-reviewed articles (Jang et al., 2019 doi: 10.1126/scitranslmed.aax5516; Foster et al., 2019 doi: 10.1007/s00204-019-02427-4). However, we revised the text in the manuscript to make this point even clearer.

2) *The authors did respond to my question about trovafloxacin-induced injury and cytokine generation by measuring both IL-6 & TNF-alpha. However, they noted no change and say that "...an additional inflammatory stimulus may be required..." My points to raise here are that the cytokine data are not shown-are they already elevated, suggesting Kupffer cell activation basally? Also, what is the "additional inflammatory stimulus"? Have the authors considered a larger cytokine panel or perhaps soluble CD163 or soluble mannose receptor?*

The levels of these cytokines that we detected in the basal channel of the control vehicle-treated Liver-Chips from all 3 donors were 300-400 pg/mL for IL-6 and 5-25 pg/mL for TNF-alpha. These levels are indicative of non-activated cells based on prior published studies investigating *in vitro* co-culture of hepatocytes and Kupffer cells (Rose et al., 2016; doi:10.1016/S0022-3549(15)00192-6). In addition, we did not detect any concentration-related change of these levels following administration of either trovafloxacin or levofloxacin. Activation of Kupffer cells typically occurs following endocytosis of foreign material, particularly bacterial endotoxins (Kolios et al., 2006 doi: 10.3748/wjg.v12.i46.7413), which is why we stated that an additional inflammatory stimulus would be required to see elevated levels of IL-6 and/or TNF-alpha. We did not include any additional inflammatory stimuli because this was not the focus of the present study. We have revised the text to reflect these findings and removed the statement about including an additional inflammatory stimulus to avoid confusion.

Use of a larger cytokine panel or measurement of the other biomarkers suggested could be useful in future studies where more specific questions relating to the inflammatory response and mechanism of drug toxicity are being explored. However, these assessments are beyond the scope of the present study which focuses specifically on the ability of human Liver-Chips to predict DILI compared to historical data from animal studies.

Reviewer #3 (Remarks to the Author):

The manuscript has improved significantly and I appreciate the transparency of the economic analysis.

The economic section of the discussion could benefit from a couple of reflections:

A) Uncertainty: Despite of including now confidence intervals and a sensitivity analysis, there is still a slight overselling of the productivity gain, which is somewhat unacademic. One important aspect is also economic and regulatory readiness that may effect realization of economic potential.

- In response to these concerns, we have made significant changes to the wording throughout the manuscript to make the style more academic. We also now include some of the modeling assumptions in the main body of the manuscript that were previously only apparent from the supplementary materials to make these points clearer for readers.
- We specifically emphasized through the new edits that the financial estimates (e.g., \$3B annually) are approximations that serve to illustrate significant potential for economic benefit – and that their magnitude seems robust in the face of input parameters and modeling assumptions – rather than precise numbers.
- We similarly clarified that the financial estimates speak to a scenario of an eventual broad adoption of the technology, rather than an instant gain. This is meant to address the Reviewer's concern about "economic readiness" mitigating the potential benefit. It will take some time for the initial investment in Organ-Chip technology to bear fruit and for the industry to reach a new steady state in terms of R&D productivity. In other words, the \$3B is now more clearly positioned in the manuscript as a potential financial reward for adopting the technology alongside the potential improvements in patient safety and reduction in animal use, as overall arguments for accelerating the technology's adoption. We now clarify these points in the revised manuscript.
- The Reviewer mentions regulatory readiness as a potential impediment for realizing the estimated financial benefit. However, our modeling only assumes the use of the Liver-Chip in a pre-regulatory context, as part of the process of selecting which compounds to progress into the regulated clinical context. We still expect regulators to review IND packages based on traditional models, albeit for compounds for which we anticipate higher probabilities of success in the clinic due to having been assessed using Organ-Chips. Accordingly, we do not believe that regulatory readiness would augment the financial estimates. We made changes to the manuscript to provide more clarity and emphasize that we only envision Organ-Chips to be used additively together with current testing modalities in our estimates.

B) Cost: Are there any estimates of the investments needed? Will there be an increase in direct cost?

- Our model includes the cost of Organ-Chip experiments as part of the NPV uplift calculation. We elected to estimate Organ-Chip expenses as they would cost if users accessed the technology through CRO; we did this because CRO pricing encompasses direct costs, indirect costs and profits, and so it should provide a top-end estimate of the expenditure. We expect the costs to be lower for a pharmaceutical company that elects to purchase their own instrumentation and apply their own employees' labor to the

testing, which would result in an even greater financial benefit than our estimates suggest. We now clarify this point in the text.

- We made the fact that we have considered the added cost of Organ-Chip experiments clearer through our edits to the manuscript. We also clarified that we have taken another additional cost into account: the increased cost of clinical trials given the increased success of compounds in the clinical (e.g., that more compounds would reach Phase III trials due to the reduction in DILI-related failures in Phase I).

C) Capacity: Will the capacity limits of the R&D pipeline naturally constrain the productivity gain? Especially for rare cancers, recruiting relevant patients for clinical trials is a massive constraint.

- For the case of using the Liver-Chip for DILI prediction, we estimate the aggregate clinical development success rate to increase only by around 4%. We view this as a modest change in clinical trial volume, and thus we expect it to introduce minimal additional burden on the recruitment of patients into clinical trials or other limits on the R&D pipeline's capacity.
- In the case of the five Organ-Chip toxicology panel (which led to the estimated \$24B in annual value generation), the model estimates that the number of Phase III clinical trials could increase by approximately 35%. Although this represents a more significant increase than in the DILI-only case, this increase in trial capacity and patient recruitment needs should be readily manageable for most (non-rare) indications. In turn, the number of compounds targeting rare diseases represented less than 10% of all compounds in clinical trials in 2020 (based on reference 74 in the manuscript); accordingly, even if the model's assumptions were to break entirely for compounds associated with rare diseases (i.e. these disease areas would see zero productivity gain), the estimated annual value generated by the technology would remain above \$20B – i.e. qualitatively in line with the listed estimate.
- That said, the edited version of the manuscript now softens the language used to describe the financial predictions to emphasize that they should be taken as estimates that illustrate significant financial potential.

REVIEWERS' COMMENTS:

Reviewer #2 (Remarks to the Author):

I appreciate the time the authors have taken to respond to the critiques of their manuscript. My point about measuring CYP protein activity is related to the fact that interindividual variability in donors is well documented. To this point, vendors will include CYP enzyme induction/inhibition as part of their QC. I still think it would be useful to have seen some basic characterization if this paper is to be cited as one which qualifies this platform e.g., midazolam hydroxylation activity. But, will defer to the editor on this decision.

Reviewer #3 (Remarks to the Author):

Thank you with your responses, I find the assumptions well justified.